# A dynamic neural resource model bridges sensory and working memory

Ivan Tomić[1,2]*, Paul M Bays[1]

[1]Department of Psychology, University of Cambridge, Cambridge, United Kingdom; [2]Department of Psychology, Faculty of Humanities and Social Sciences, University of Zagreb, Zagreb, Croatia

**Abstract** Probing memory of a complex visual image within a few hundred milliseconds after its disappearance reveals significantly greater fidelity of recall than if the probe is delayed by as little as a second. Classically interpreted, the former taps into a detailed but rapidly decaying visual sensory or 'iconic' memory (IM), while the latter relies on capacity-limited but comparatively stable visual working memory (VWM). While iconic decay and VWM capacity have been extensively studied independently, currently no single framework quantitatively accounts for the dynamics of memory fidelity over these time scales. Here, we extend a stationary neural population model of VWM with a temporal dimension, incorporating rapid sensory-driven accumulation of activity encoding each visual feature in memory, and a slower accumulation of internal error that causes memorized features to randomly drift over time. Instead of facilitating read-out from an independent sensory store, an early cue benefits recall by lifting the effective limit on VWM signal strength imposed when multiple items compete for representation, allowing memory for the cued item to be supplemented with information from the decaying sensory trace. Empirical measurements of human recall dynamics validate these predictions while excluding alternative model architectures. A key conclusion is that differences in capacity classically thought to distinguish IM and VWM are in fact contingent upon a single resource-limited WM store.

## eLife assessment

This study presents **important** insights into the dynamical process whereby sensory information is converted from stimulus-driven activity to a working memory representation from which the information can be recalled later. The evidence supporting the claims is **convincing**, using detailed fits and model-comparison techniques applied to new and existing psychophysical data sets to evaluate a wide variety of potential mechanisms. The overall conclusion, that iconic memory and working memory are not distinct mechanisms but rather two slightly different regimes of the same circuitry, will be of interest to neuroscientists and psychologists studying sensory systems and/or working memory.

## Introduction

Keeping relevant information in an easily accessible state is vital for adaptive behavior in dynamic environments. In the primate visual system, this requirement is met by visual working memory (VWM), the capacity to actively maintain visual information from milliseconds to seconds after a stimulus disappears from view (*D'Esposito and Postle, 2015*; *Pasternak and Greenlee, 2005*; *Ma et al., 2014*; *Bays et al., 2024*). While the contents of VWM are frequently updated to reflect changes in the environment and in behavioral priorities, the visual processing hierarchy itself introduces additional layers of dynamism (*Barlow, 1981*; *Van Essen et al., 1992*). The fidelity of representations therefore evolves

*For correspondence:
ivn.tomic@gmail.com

from the moment VWM starts accumulating evidence (*Brunton et al., 2013*; *Gold and Shadlen, 2007*) throughout the maintenance period until the information is used for action (*Schneegans and Bays, 2018*; *Panichello et al., 2019*; *van Ede et al., 2019*).

Nonetheless, within most theoretical frameworks, VWM is treated as a stationary process whereby representations are measured and modeled as fixed states of the system. One such model of WM is based on principles of neural population coding (*Bays, 2014*; *Schneegans et al., 2020*). In the Neural Resource model, visual information is encoded in the activity of a population of noisy feature-selective neurons (*Ma et al., 2006*; *Pouget et al., 2000*). The spiking activity of the neural population is constrained by normalization (*Carandini and Heeger, 2011*; *Bays, 2015*), such that the total activity is fixed but flexibly distributed between memoranda, implementing a form of limited memory resource. At retrieval, encoded stimulus values are reconstructed from the noisy spiking activity. This model has provided a quantitative account of patterns of recall error across a range of tasks and stimulus dimensions (*Tomić and Bays, 2024*; *Bays and Taylor, 2018*; *Schneegans and Bays, 2017*; *Bays, 2016a*; *Tomić and Bays, 2018*). However, despite its grounding in principles of neural coding, the basic architecture of the model lacks a temporal dimension to describe the dynamics of memory representations during encoding and maintenance.

Research on prolonged memory maintenance has demonstrated that the precision of stored representations gradually deteriorates over time (e.g. *Pertzov et al., 2017*; *Rademaker et al., 2018*). Computational models attempting to account for these dynamics have often relied on principles of diffusion within an attractor network. In such a network, information is maintained in a sustained pattern of activity, which can be visualized as a 'bump' of activity centered on the stored value. Over time, the bump diffuses along the feature dimension due to random fluctuations in neural activity, leading to stochastic changes in the encoded feature value and a gradual loss of information (*Burak and Fiete, 2012*; *Wimmer et al., 2014*). Critically, the neural code diffuses without decay in signal strength. A growing body of empirical support, both at the behavioral (*Schneegans and Bays, 2018*) and neural level (*Lim et al., 2019*; *Wolff et al., 2020*), identifies diffusion as a key mechanism of memory deterioration.

In contrast to such gradual deterioration over longer retention intervals, studies that probed memory within a few hundred milliseconds of stimulus offset revealed a precipitous decrease in memory fidelity immediately after a stimulus disappears (*Di Lollo and Dixon, 1988*; *Sperling, 1960*; *Bradley and Pearson, 2012*; *Pratte, 2018*). This early superior recall was attributed to a high-capacity but short-lived form of storage termed iconic memory (IM) (*Neisser, 1967*). An implicit assumption has often been that the behavioral advantage of early cues derives from reading out information directly from IM and circumventing capacity limitations imposed by VWM, however, this idea has not been formally modeled or tested. At the neural level, IM is thought to be supported by a brief period of decaying neural activity in early visual areas following the response elicited by the visible stimulus (*Priebe et al., 2002*; *Rolls and Tovee, 1994*; *Teeuwen et al., 2021*; *van Kerkoerle et al., 2017*). In contrast to later memory dynamics arising due to noise accumulation, early changes in memory fidelity were supported by modulation of the neural signal strength. However, little is known about the read-out of this sensory memory buffer.

Finally, memory fidelity changes during encoding while the evidence is extracted from the visible stimulus. Previous studies revealed that longer stimulus exposures have a favorable effect on the subsequent recall, but that this effect is modulated by the number of simultaneously encoded objects (*Bays et al., 2011*; *Shibuya and Bundesen, 1988*; *Vogel et al., 2006*), providing evidence for a processing or encoding limitation of VWM. As stimulus presentation duration increases, more information may be extracted from the sensory signal into VWM, increasing the fidelity of the representation. Critically, with prolonged exposure, VWM fidelity approaches a stable level that depends on the number of encoded items, suggesting that a ceiling is imposed on evidence accumulation by a shared limit on VWM resources. However, a computational framework describing information accumulation from sensory areas into VWM is lacking, and the observed encoding limit may reflect dynamics in sensory areas registering visible objects as well as VWM accumulating this sensory evidence.

Here, we investigated the temporal dynamics in the fidelity of VWM from information encoding until its recall. To map human recall fidelity to the time domain, we conducted psychophysical experiments in which we probed memory representations at different time points relative to stimulus onset and offset while simultaneously manipulating set size. To isolate memory dynamics due to changes

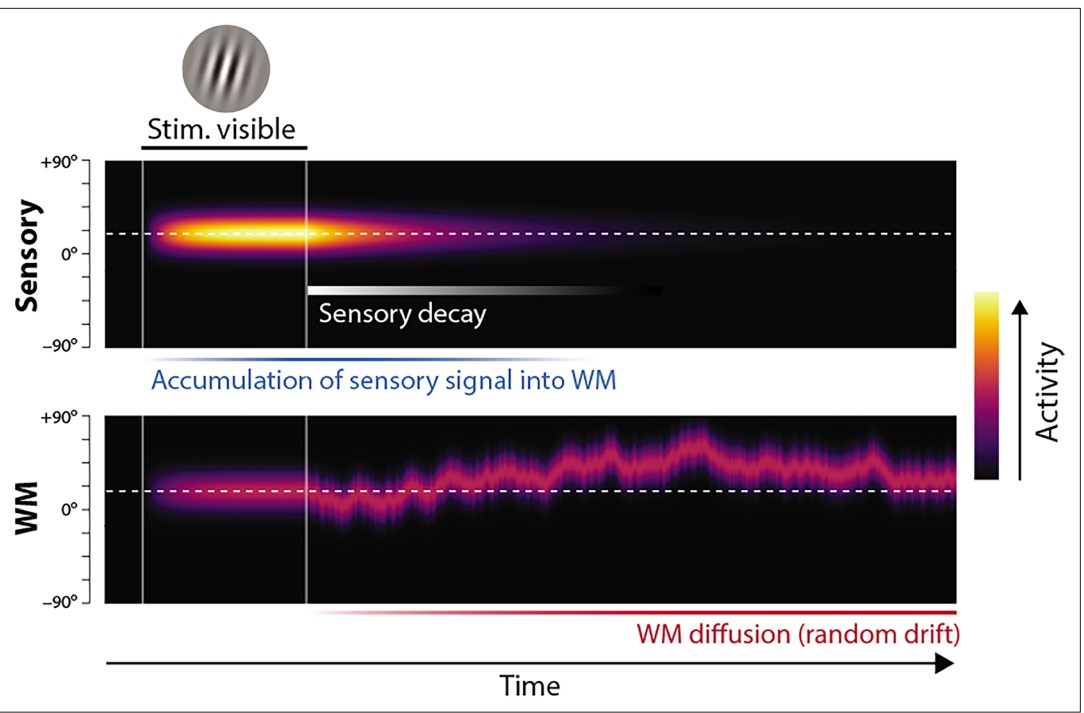

**Figure 1.** Proposed neural population dynamics for encoding a single orientation into visual working memory (VWM) and maintaining it over a delay. Top: Stimulus onset is followed by a ramping increase in activity (indicated by color) of sensory neurons whose tuning (indicated on *y* axis) matches the stimulus orientation. Following stimulus offset, this sensory signal rapidly decays. The sensory signal, including its decaying post-stimulus component, provides input into VWM. Bottom: At stimulus onset, the VWM population begins to accumulate activity from the sensory population. This accumulation saturates at a maximum amplitude determined by global normalization. As the sensory activity decays, the activity in the VWM population is maintained at a constant amplitude, but accumulation of random errors causes the activity bump to diffuse along the feature dimension (*y* axis) over time, changing the orientation represented by the population. At recall, when the VWM population activity is decoded, accuracy of the recall estimate depends on both the orientation represented (center of the activity bump) and the fidelity with which it can be retrieved (determined by activity amplitude).

in the representational signal, we advanced an analogue reproduction task with a novel response method specifically adapted to minimize the time cost of motor (i.e. response) processes and capture the momentary state of memory representations. This allowed us to precisely measure the time course of fidelity dynamics during representation formation (i.e. encoding) and retention (i.e. maintenance). A major conclusion is that the enhanced precision seen at very brief retention intervals depends on integration of information from the sensory store into VWM following the cue, with the result that retrieval from IM of even the simplest stimulus is subject to the temporal and capacity limitations of WM.

To explain the neural computations underlying the observed time courses, we devised a comprehensive neural model of memory dynamics whose core architecture is rooted in the Neural Resource model of VWM (**Bays, 2014**; **Schneegans et al., 2020**). The Dynamic Neural Resource (DyNR) model assumes that changes in memory fidelity reflect temporal dynamics in the sensory population registering the stimuli and from signal and noise accumulation processes of resource-limited VWM (**Figure 1**). In particular, the model prescribes how time-dependent gain control mechanisms in sensory areas produce a smooth neural response following abrupt changes in stimulus presence. As this sensory signal provides feedforward input to VWM, the dynamics in VWM activity in the temporal vicinity of stimulus presentation (i.e. onset and offset) strongly reflect not only limits in VWM, but also the dynamics of the sensory signal. Finally, once accumulated into VWM, the neural signal is subject to perturbations due to noise accumulation, resulting in degradation of internal representations with time. The DyNR model accurately reproduced the detailed empirical patterns of human recall errors in the psychophysical experiments. Based on these results, we argue that changes in memory fidelity on short time scales reflect dynamics in the gain or signal strength in neural populations representing

the stimulus, while changes on longer time scales are dominated by corruption of the representation by accumulated noise.

## Dynamic Neural Resource (DyNR) model

The DyNR model generalizes an established neural population account of VWM, originally proposed by *Bays (2014)*, and inspired by similar models of attention and perceptual decision-making (*Jazayeri and Movshon, 2006*; *Ohshiro et al., 2011*; *Reynolds and Heeger, 2009*). In the original model, memorization and recall of visual stimuli is achieved by encoding and decoding of spiking activity in idealized feature-tuned neurons. The limited capacity of VWM to hold multiple object features simultaneously is reproduced by a global divisive normalization that constrains total spiking activity, implementing a continuous memory resource (*Carandini and Heeger, 2011*; *Bays, 2014*). The DyNR model (illustrated in *Figure 1*) extends this stationary encoding-decoding model with a temporal dimension. First, to capture encoding dynamics, stimulus information enters the VWM population (*Figure 1*, bottom) indirectly, by accumulation of neural signal from a separate sensory population (top), which receives the visual input. The signal strength in the VWM population at any point in time jointly depends on the history of the signal in the sensory population and the number of features competing for representation in VWM. Once the sensory signal is gone, the VWM signal is maintained at its maximum attained amplitude, but the stimulus value encoded by the signal gradually diffuses due to accumulation of random noise. Recall error depends on both the stimulus value represented at the time of retrieval (*what* is encoded) and the signal amplitude at that time, read out in the form of spikes (*how precisely* it can be decoded).

## Dynamics of sensory signal strength

To model the temporal dynamics of human memory fidelity, we begin by defining computations of the sensory system registering the incoming signal. A particularly important computation is temporal filtering – a property of neurons to respond more sensitively to specific temporal patterns in stimuli. To model the signal represented in the cortical sensory level, we assume that the sensory response to a stimulus presentation of fixed duration (described as a step function in visual input amplitude, *Figure 2A and B*, left) is controlled by a monophasic temporal filter having a low-pass frequency response (*Hess and Snowden, 1992*). This choice is a natural one since it is consistent with electro-physiological studies demonstrating that a large range of temporal frequencies registered by the retina and LGN (*Derrington and Lennie, 1984*; *Lee et al., 1989*) is attenuated at higher frequencies before the signal enters the primary visual cortex (*Hawken et al., 1996*). Passing the stimulus through such a temporal filter attenuates the neural response to fast transients in the signal, and thereby produces a smooth rise and decay of neural activity in response to a uniform input signal (*Figure 2C*). In particular, we assume that the activity of the sensory population after stimuli onset and offset changes exponentially toward the maximum sensory activity and baseline activity, respectively. The choice of the filter's temporal response characteristics (i.e. its time constant) fully defines dynamics in the sensory population activity and controls the signal projected toward higher areas. The available physiological evidence suggests the temporal properties of the rising and decaying neural response are not symmetric (*Müller et al., 2001*; *Oram and Perrett, 1992*; *Ringach et al., 2003*). In particular, the neural response typically reaches the maximum activity after the onset faster than it reaches the baseline activity after the offset. Consistent with this, we allowed the sensory signal to decay at a different rate than the rising rate. The temporal dynamics in sensory population firing activity in response to a fixed input signal of duration $t_{\text{offset}}$ is then given by:

$$
\dot{\gamma}_{\text{s}}(t) = \begin{cases} (\tilde{\gamma}_{\text{s}} - \gamma_{\text{s}}(t))/\tau_{\text{rise}} & \text{for} \quad t \leq t_{\text{offset}} \\ -\gamma_{\text{s}}(t)/\tau_{\text{decay}} & \text{for} \quad t > t_{\text{offset}} \end{cases}
\tag{1}
$$

where $\tilde{\gamma}_{\text{s}}$ is the maximum sensory signal, $\tau_{\text{rise}}$ and $\tau_{\text{decay}}$ are rising and decaying time constants of the temporal filter, respectively.

The temporal properties of the sensory response have been shown to depend on the physical characteristics of stimuli, such as contrast and location (*Müller et al., 2001*; *Sit et al., 2009*). Similarly, previous work has demonstrated that the decaying component of the sensory response is strongly influenced by the engagement of the sensory population after stimuli offset (e.g. *Rolls and Tovee,*

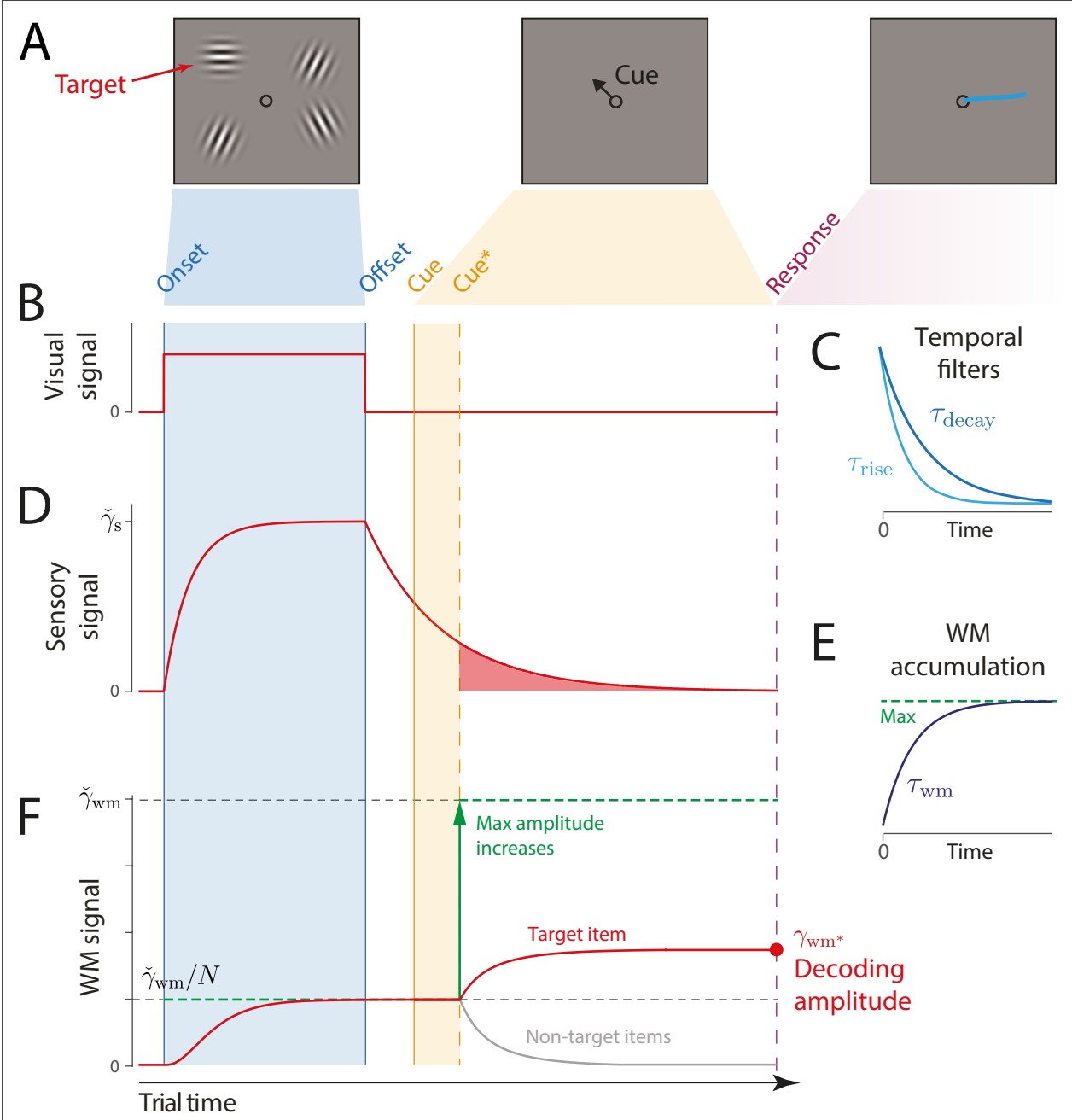

**Figure 2.** Schematic of signal amplitudes in the dynamic neural resource (DyNR) model during a cued recall trial. (**A**) Observers are presented with a memory array (left), followed after a blank delay (not shown) by an arrow cue (center) indicating the location of one item (the target) whose remembered orientation should immediately be reported (right). (**B**) The amplitude of the visual input associated with each item is modeled as a step function (left). The sensory response (**D**) is modeled as a low-pass filtering of the stimulus input, with different time constants for rise and decay (**C**). (**F**) Amplitude of the working memory signal reflects a saturating accumulation of activity from the sensory population (illustrated in **E**). Beginning with stimulus onset, activity associated with each item is accumulated from the sensory population into the visual working memory (VWM) population, approaching an upper bound (green dashed line) that reflects a total activity limit shared between the $N$ items in memory. Once the cue has been presented (solid orange line) and processed (dashed orange line), uncued items can be dropped from VWM, raising the ceiling on activity available to represent the cued item (green arrow). This allows more information about the cued item to be accumulated from the decaying sensory trace (equivalent to the red shaded area in D). Response variability depends on the asymptotic VWM signal amplitude available for decoding (red circle) combined with the accumulated effects of diffusion (see text).

*1994*). In particular, a new input signal, e.g., a backward noise mask, curtails ongoing activity related to the previous stimulus, resulting in a faster decay of activity compared to the unmasked post-stimulus period (*Kovács et al., 1995*). Consistent with this, here we assume that the backward mask operates by interrupting ongoing sensory processing of stimuli, limiting the access to the sensory signal (Figure 5) (cf. integration mask) (*Turvey, 1973*).

## Dynamics of VWM signal strength

The information registered by the sensory system is subsequently accumulated into a VWM population capable of maintaining activity in the absence of further input (e.g. by self-excitation, see *Aksay et al., 2001*; *Wimmer et al., 2014*; *Compte et al., 2000*; although only the resulting dynamics are modeled here). The total activity of the VWM neural population is normalized, implementing a limited resource shared out between memory items (*Bays, 2014*; *Schneegans et al., 2020*). Consequently, if the stimuli are presented for long enough, the evidence accumulated from the sensory signal into VWM will saturate at a level that reflects the total number of stimuli represented (*Figure 2D*). The dynamics in VWM population activity are given by:

$$\dot{\gamma}_{\text{wm}}(t) = \gamma_{\text{s}}(t)(\tilde{\gamma}_{\text{wm}}/M(t) - \gamma_{\text{wm}}(t))/\tau_{\text{wm}} \tag{2}$$

where $\tilde{\gamma}_{\text{wm}}$ is the maximum VWM signal amplitude, $M(t)$ is the number of items represented in VWM at time $t$, $\tau_{\text{wm}}$ is the time constant of accumulation into VWM.

A common assumption of VWM models is that the strength of the representational signal remains stable after encoding from a visible stimulus. This stationary view has been reinforced by typically measuring VWM sufficiently long after the stimulus disappears (~1 s) and at a single time point. In contrast, work on IM demonstrated that recall fidelity in a brief period after stimulus offset typically surpasses and then precipitously decays toward VWM fidelity level (*Coltheart, 1980*). Consistent with that, we consider how the normalized representational signal in VWM formed during encoding can be boosted in the absence of the physical stimulus. In particular, we assume a representation stored in VWM can be strengthened as long as the sensory population provides feedforward input and VWM activity is not saturated at the normalized level. Such a scenario can be achieved by cueing an item for recall in the temporal vicinity of stimulus offset, i.e., before sensory activity decays to zero. By cueing an item for recall, the remaining contents of VWM becomes obsolete and can be removed from memory (*Oberauer, 2018*). In the model,

$$M(t) = \begin{cases} N & \text{for} \quad t \leq t_{\text{cue}*} \\ 1 & \text{for} \quad t > t_{\text{cue}*} \end{cases} \tag{3}$$

where $t_{\text{cue}*}$ is the time when the item is identified for a recall and the read-out of stimulus value begins. This 'demounting' of resource from uncued items makes it available for storing additional information about the cued item, which is extracted from the residual sensory representation, increasing the representation fidelity beyond that granted by equal distribution of neural signal between items. Critically, as sensory information quickly decays, there will be less signal remaining to supplement the VWM representation of a cued item if the cue is delivered later, and at the longest cue intervals the cue will confer no advantage over the fidelity attained when all items compete equally for VWM representation (*Figure 2D*). We note that removal of uncued items cannot occur until the cue has been processed to the point of identifying 1 of the $N$ items in the memory array. We follow *Hick, 1952*, in modeling this cue processing time as logarithmic in the number of alternatives:

$$t_{\text{cue}*} = t_{\text{cue}} + b \log_2(N) \tag{4}$$

where $b$ is a scaling parameter. Previous work demonstrated that estimation of temporal dynamics in attention and memory could be confounded with the time needed to interpret the cue and start acting on it (*Shih and Sperling, 2002*). This is especially significant when trying to accurately capture quickly changing processes, such as decay of the sensory residual. Although the cue processing time likely fluctuates on a trial-by-trial basis due to changes in, e.g., attention, arousal, or motivation, here we focus on the influence of set size arising from a limited information processing capacity.

## Diffusion of VWM encoded values

So far we have described only changes in the strength of the neural signal encoding features in memory. However, feature representations maintained over time in neural activity will accumulate noise in the absence of external input. We model this process of noise-driven diffusion as Brownian motion in feature space throughout the retention interval (*Figure 1*), contributing to variability in the decoded feature value (*Burak and Fiete, 2012*; *Schneegans and Bays, 2018*). The resulting variability is described by a wrapped normal distribution with variance $\sigma^2$ that increases linearly with time from stimulus offset, so that at time $t$ the encoded feature corresponding to a true stimulus feature $\theta$ is:

$$\theta(t) \sim \mathcal{WN}(\theta, \sigma^2(t)) \tag{5}$$

$$\sigma^2(t) = (t - t_{\text{offset}})\dot{\sigma}^2_{\text{diff}} \tag{6}$$

where $\dot{\sigma}^2_{\text{diff}}$ specifies the base diffusion rate. While the fast decay of sensory activity after stimuli offset accounts for early dynamics in VWM fidelity, diffusion becomes prominent over longer delays, accounting for more gradual deterioration of precision with time.

Such a diffusion account has support in the available neural evidence as well as in theoretical work. At the neural level, an electrophysiological study in monkeys performing a spatial WM task demonstrated that shifts of neural tuning curves during a memory delay predicted behavioral response errors (*Wimmer et al., 2014*). A similar finding was observed in humans where drift in the fMRI activity patterns relative to the target predicted errors in an orientation discrimination task (*Lim et al., 2019*). At a theoretical level, continuous attractor models explain diffusion as a consequence of neural variability in networks where excitatory and inhibitory connections constrain population activity to a subspace or manifold corresponding to the encoded feature space (*Burak and Fiete, 2012*; *Bouchacourt and Buschman, 2019*; *Compte et al., 2000*).

## Retrieval

To model the process that leads to a response we first consider that in some trials observers may erroneously identify a non-target item as being cued. Previous work indicates these 'swap' errors occur due to uncertainty in memory for the cue features of the stimuli, in this case their locations (*Schneegans and Bays, 2017*; *McMaster et al., 2022*). We assume that changes in variability in the cue features mirror those of the memory features, leading swap frequency to decrease exponentially as a function of presentation duration and increase linearly with retention interval (*Appendix 2—figure 1*):

$$p_{\text{swap}} = (N-1)\left[\left(\frac{1}{N} - r_{\text{spatial}}t_{\text{cue}*}\right)e^{\frac{-t_{\text{offset}}}{\tau_{\text{spatial}}}} + r_{\text{spatial}}t_{\text{cue}*}\right] \tag{7}$$

where $\tau_{\text{spatial}}$ is the time constant related to presentation duration, and $r_{\text{spatial}}$ is the rate constant related to the retention interval.

If $\theta$ is the true feature value of the item identified as the target (i.e. the cued item with probability $1 - p_{\text{swap}}$, a randomly selected non-cued item with probability $p_{\text{swap}}$), then due to diffusion (*Equation 5*) the value encoded in the VWM population at the time of retrieval is given by:

$$\theta^* \sim \mathcal{WN}(\theta, \sigma^2(t_{\text{cue}*})) \tag{8}$$

We model retrieval as estimation of $\theta^*$ based on spiking activity in the VWM population that encodes the selected item. For this purpose we assume an idealized set of tuning functions, where the mean response of neuron $i$ encoding orientation $\theta$ with population gain $\gamma$ is described by:

$$f_i(\theta, \gamma) = \frac{\gamma}{n}\exp(\kappa(\cos(\theta - \varphi_i) - 1)) \tag{9}$$

where $n$ is the number of neurons, and $\kappa$ determines the tuning width. The preferred orientations of the neurons, $\varphi_i$, are evenly distributed throughout the circular space to provide uniform coverage. The spike count produced by each neuron is drawn from a Poisson distribution,

$$r_i \sim \text{Poisson}(f_i(\theta^*, \gamma_{\text{wm}*})) \tag{10}$$

and the decoded orientation estimate is obtained by ML estimation based on the spike counts:

$$\hat{\theta} = \arg\max_{\theta} \; p(\mathbf{r}|\theta). \tag{11}$$

## Additional assumptions

To fit the model to behavioral data, we make several further simplifying assumptions. We assume that the exponential decay of the sensory signal is rapid enough that there is effectively no information remaining by the time the VWM population is decoded to generate a response. This allows us to approximate the VWM activity at the time of decoding by the asymptotic VWM activity were the sensory decay to continue indefinitely:

$$\gamma_{\mathrm{wm}*} \approx \gamma_{\mathrm{wm}}(\infty) \tag{12}$$

Next, we identify diffusion in the encoded value at the time of retrieval with diffusion at the time of target item identification, justifying the use of $t_{\mathrm{cue}*}$ in **Equation 8**. We reason that the rate of diffusion

**Table 1.** Dynamic neural resource (DyNR) model parameters (1–9) and other variables (10–24) used in model description.

| No. | Parameter/variable | Description |
|---|---|---|
| 1 | $\tilde{\gamma}_{\mathrm{wm}}$ | Maximum VWM signal amplitude |
| 2 | $\kappa$ | Tuning curve width |
| 3 | $\tau_{\mathrm{rise}}$ | Rise constant of the sensory temporal filter |
| 4 | $\tau_{\mathrm{decay}}$ | Decay constant of the sensory temporal filter |
| 5 | $\tau_{\mathrm{wm}}$ | Time constant of accumulation into VWM |
| 6 | $\dot{\sigma}^2_{\mathrm{diff}}$ | Base diffusion rate |
| 7 | $\tau_{\mathrm{spatial}}$ | Time constant for spatial encoding |
| 8 | $r_{\mathrm{spatial}}$ | Rate constant for spatial diffusion |
| 9 | $b$ | Scaling parameter for Hick's law |
| 10 | $t$ | Time, relative to stimulus onset ($t = 0$) |
| 11 | $t_{\mathrm{offset}}$ | Time of stimulus offset |
| 12 | $t_{\mathrm{cue}}$ | Time of cue onset |
| 13 | $t_{\mathrm{cue}*}$ | Time an item is identified for report |
| 14 | $N$ | Number of items in stimulus array |
| 15 | $M(t)$ | Number of items in memory at time $t$ |
| 16 | $\tilde{\gamma}_{\mathrm{s}}$ | Maximum sensory signal amplitude |
| 17 | $\gamma_{\mathrm{s}}(t)$ | Sensory signal amplitude at time $t$ |
| 18 | $\gamma_{\mathrm{wm}}(t)$ | VWM signal amplitude at time $t$ |
| 19 | $\gamma_{\mathrm{wm}*}$ | VWM signal amplitude at the time of decoding |
| 20 | $\sigma^2(t)$ | Accumulated diffusion at time $t$ |
| 21 | $n$ | Number of neurons |
| 22 | $\theta$ | True stimulus feature value |
| 23 | $\theta^*$ | Encoded stimulus feature value at the time of decoding |
| 24 | $\hat{\theta}$ | Decoded stimulus feature value |

is slow enough relative to the rate of sensory decay, that any additional diffusion in the brief period of post-cue sensory accumulation is negligible.

In Experiment 1 (see below), a task with a fixed 200 ms exposure period, we assume that the initial encoding of all items into VWM is complete by the time of stimulus offset, i.e., that VWM activity at this time can be approximated by its asymptotic level reflecting normalization:

$$\gamma_{\mathrm{wm}}(t_{\mathrm{offset}}) \approx \tilde{\gamma}_{\mathrm{wm}}/N \qquad (13)$$

Finally, in the condition of Experiment 1 where memory array and cue are presented simultaneously, we assume that only the cued feature is encoded in VWM, reaching the maximum amplitude, $\tilde{\gamma}_{\mathrm{wm}}$, irrespective of set size. Maximum likelihood (ML) fits were obtained via the Nelder-Mead simplex

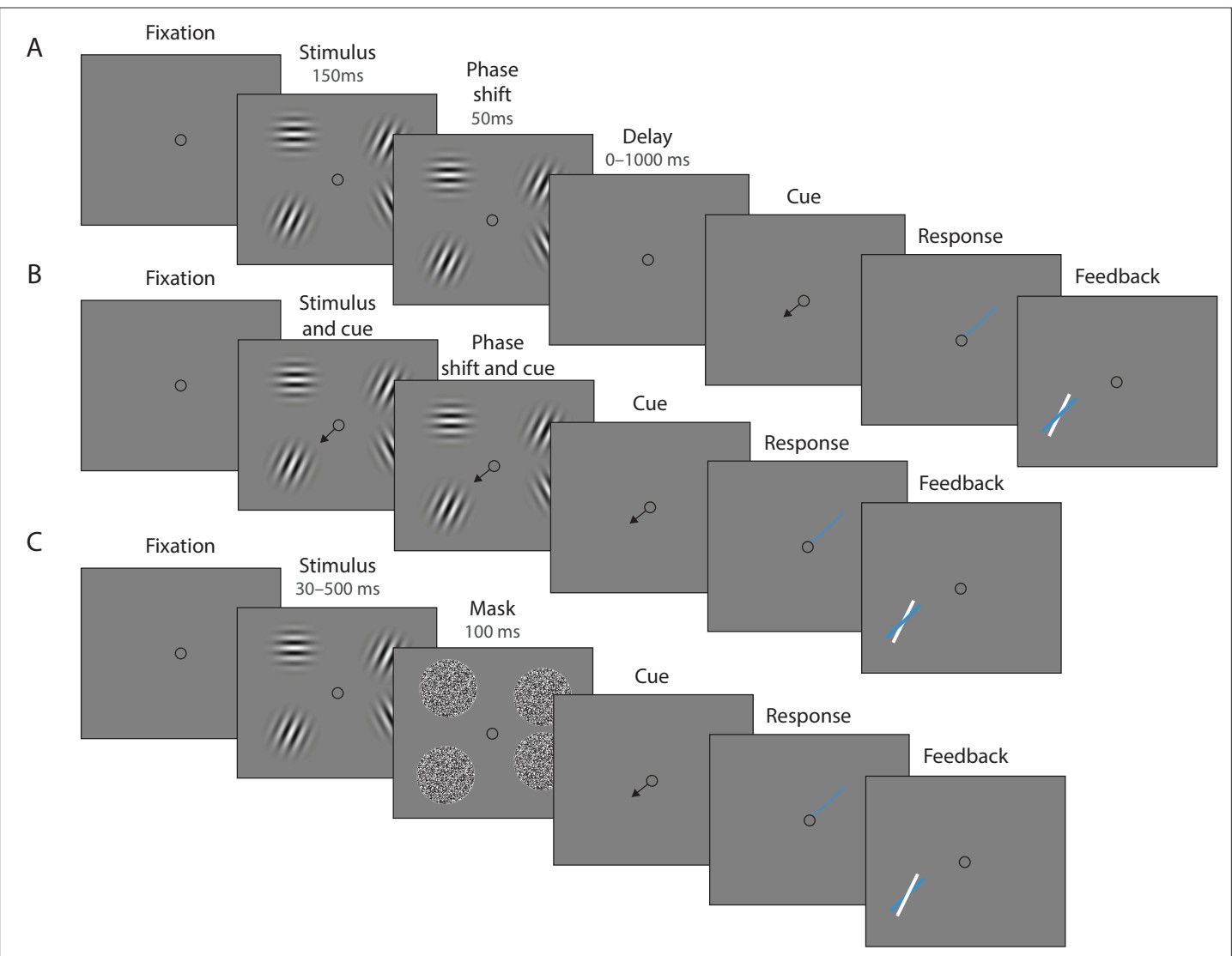

**Figure 3.** Experimental procedure. (**A**) Experiment 1. On each trial, a memory array was presented consisting of 1, 4, or 10 randomly oriented Gabor stimuli. In 50% of all trials, the stimuli underwent a change of phase and contrast toward the end of the exposure period intended to minimize retinal after-effects. After a variable delay, an arrow cue was shown pointing toward the location of one stimulus from the preceding array. Observers reported the remembered orientation of the cued stimulus by swiping their index finger on the touchpad. The response was followed by feedback showing the true orientation. (**B**) In a proportion of trials, the cue was presented simultaneously with the stimuli. (**C**) Experiment 2. On each trial a memory array consisting of 1, 4, or 10 randomly oriented Gabor was presented for a variable duration, and followed by a white noise flickering mask. The mask was replaced by an arrow cue pointing toward the location of one stimulus from the preceding array. Observers reported remembered its orientation and received feedback as in Experiment 1. Stimuli are not drawn to scale.

method (function *fminsearch* in Matlab). All parameters and variables used to describe the DyNR model are listed in *Table 1*.

## Overview of experiments

We tested predictions of the DyNR model against empirical data collected in continuous report tasks. In Experiment 1 (*Figure 3A and B*), observers were presented with an array of oriented stimuli for a fixed duration followed after a variable delay by a visual cue identifying one of the preceding stimuli whose orientation should be reported. This experiment was designed to investigate the contribution of decaying sensory representations following stimulus offset to the dynamics of recall fidelity. Experiment 2 (*Figure 3C*) was aimed at expanding the results of the first experiment to now also assess the accumulation of information during the time the stimuli were visible. In this case, the exposure duration was varied while the delay before the visual cue was held constant. In both experiments we varied the number of stimuli in the array (set size) to assess capacity limitations affecting encoding and maintenance.

To provide additional validation of the DyNR model, we also tested its predictions against data from a previously published continuous report experiment (Experiment 1 in *Bays, 2014*) and one additional dataset collected as part of a separate study (*Tomić et al., 2024*). A detailed description of all experiments is provided in the Methods section.

## Results

### Experiment 1: Delay duration

In Experiment 1, we evaluated the time course of VWM fidelity over brief memory intervals. Previous work has demonstrated that immediately after a stimulus physically disappears, its representation briefly persists in the sensory system in the form of residual neural activity (*Teeuwen et al., 2021*). Accumulation of this lingering sensory activity into VWM could enable superior recall of information (*Coltheart, 1980*) within the constraints of a finite VWM resource that strongly limits representational fidelity (*Ma et al., 2014*). To describe these dynamics, we examined human recall of orientation stimuli presented in arrays of varying sizes and probed after a variable delay ranging from 0 ms to 1000 ms. Here, we focus on an experimental condition in which retinal afterimages were suppressed by a phase

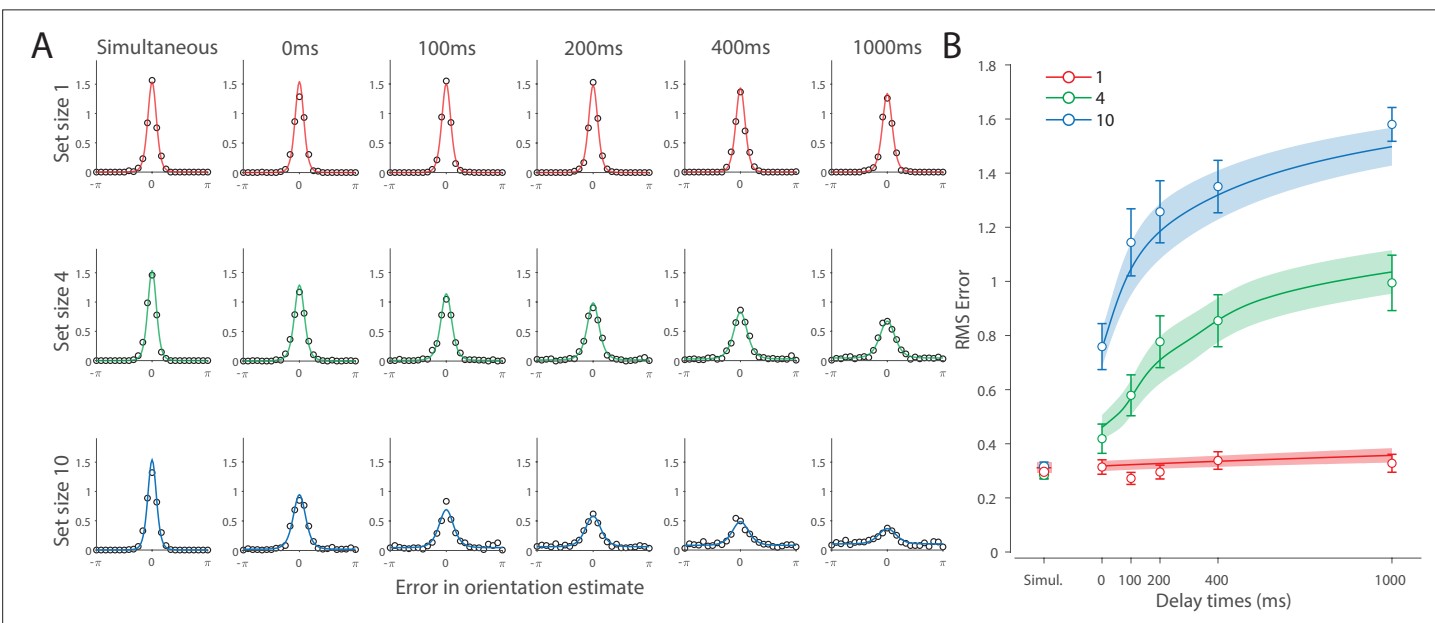

**Figure 4.** Experiment 1 data and model fits show the consequences of varying set size and delay duration on working memory (WM) reproduction error. (**A**) Empirical recall error distributions (black circles) and the dynamic neural resource (DyNR) model fits (colored curves). Different panels correspond to different set sizes (rows) and delays (columns). (**B**) Corresponding RMS errors from experimental data (circles and error bars) and the DyNR model fits (curves and error patches). Error bars and patches indicate ±1 SEM. N = 10.

shift toward the end of stimuli presentation. Validation of this method and results from the condition without a phase shift are provided in Appendix 1.

## Experimental data

Recall error distributions and mean performance in Experiment 1 are plotted in *Figure 4A and B*. Response error (measured as RMSE) increased with both set size and delay duration. A repeated measures ANOVA revealed a significant effect of set size ($F_{(2,18)} = 117.8, \mathrm{p} < 0.001, \eta^2 = 0.44$), delay time ($F_{(5,45)} = 52, \mathrm{p} < 0.001, \eta^2 = 0.23$), and their interaction ($F_{(10,90)} = 26.7, \mathrm{p} < 0.001, \eta^2 = 0.13$) on response error. We further explored this interaction, first finding response error in the 1 item condition (red in *Figure 4*) did not change with delay ($F_{(5,45)} = 1.32, \mathrm{p} = 0.27, \eta^2 = 0.07$). This was supported by Bayesian analysis ($BF_{10} = 0.34$) which found weak to moderate evidence against modulation of 1 item recall by memory delay. In contrast, response error increased with delay for the remaining two set sizes (4 items, green; 10 items, blue; main effect: $F_{(5,45)} = 55, \mathrm{p} < 0.001, \eta^2 = 0.48$). This increase in response error consisted of an initial rapid rise (over the first 200 ms), followed by a more gradual increase as the delay between stimulus and cue increased. Next, we found a modulating effect of delay on recall for the remaining two set sizes (interaction: $F_{(5,45)} = 10.1, \mathrm{p} < 0.001, \eta^2 = 0.05$). The direct comparison revealed that the increase in response error with delay ($\Delta\mathrm{RMSE} = \mathrm{RMSE}_{1000\mathrm{ms}} - \mathrm{RMSE}_{\mathrm{Simult}}$) was greater when observers memorized more items ($t_{(9)} = 9.1, \mathrm{p} < 0.001, d = 2.88$).

One surprising result was the observed set size effect in the 0 ms delay condition ($F_{(2,18)} = 23.7, p < .001, \eta^2 = .53$) consistent with a stepwise increase in recall error with set size (pairwise comparison, $t_{(9)} \geq 2.88, p \leq .036, d \geq 0.91$, Bonferroni correction applied). Importantly, this effect was a consequence of responding based on a memory of the stimulus, since orientation reproduction was comparable across set sizes in the perceptual condition (simultaneous presentation; $F_{(2,18)} = 1.26, p = .3, \eta^2 = .04, BF_{10} = 0.47$). Previous studies have characterized IM as an effectively unlimited store, capable of holding any number of items without a consequent loss of fidelity (***Doost and Turvey, 1971***; ***Sperling, 1960***). While our modeling ultimately affirmed this conception of IM, we nonetheless show that recall of information is contingent on the number of objects concurrently in memory from the moment stimuli physically disappear (see below).

Taken together, these results provide evidence that the fidelity of stored representations changes dramatically over the first few moments after stimuli offset. We next aimed to explain the neural computations supporting these dynamics. In summary, behavioral data displayed three key characteristics we aimed to explain, all visible in *Figure 4B*. First, recall fidelity for a single item remained relatively stable across changes in delay, and was the same as perceptual fidelity. Second, recall fidelity for higher set sizes showed substantial, nonlinear temporal dynamics. Lastly, recall fidelity was contingent on the number of stored items from the moment stimuli disappeared.

## DyNR model

Curves in *Figure 4A and B* show fits of the model with ML parameters (mean ± SE: population gain $\gamma$ = 59.8 ± 3.3, tuning width $\kappa$ = 3.21 ± 0.2, sensory decay time constant $\tau_{\mathrm{decay}}$ = 0.21 ± 0.052, VWM accumulation time constant $\tau_{\mathrm{WM}}$ = 0.096 ± 0.045, cue processing constant $b$ = 0.171 s ± 0.055 s, base diffusion $\sigma^2_{\mathrm{diff}}$ = 0.03 ± 0.017, swap probability $p$ = 0.027 ± 0.009). The model provided a close fit to response error distributions (*Figure 4A*) and summary statistics (*Figure 4B*; see also *Appendix 2— figure 1* for reproduction of swap error frequencies), successfully reproducing the pattern of changes with set size and delay. In particular, the model accounted for the three key observations identified above.

First, the model predicted the near-constant recall fidelity observed for a single item across these short retention intervals. The neural signal associated with the target object at recall depends on the normalized signal in VWM at offset supplemented by the available sensory signal post-cue. The sensory signal is integrated into VWM after the cue to fill any unallocated neural resource that arose by discarding uncued items. In the case of a single item, the entirety of VWM resources are allocated to one object during encoding, so no resource is freed by the cue that would allow the signal to be further strengthened based on the decaying sensory representation.

Importantly, this prediction contradicts the classical view of direct read-out from IM, according to which representational fidelity should be enhanced with very short delays irrespective of VWM limitations (see *Alternative accounts* below for a formal test of such a model). Note that the DyNR model

nonetheless predicts some deterioration in fidelity over time even for a single item, due to noise-driven diffusion of the stored value. However, based on previous reports, we expected this process to be substantially slower and the impact on single-item precision relatively small on this (≤1 s) time scale. The fitted diffusion parameters and resulting shallow slope of fitted RMS error (red curve in *Figure 4B*) confirmed this.

Second, the neural model predicts the specific pattern of dynamics observed in trials with multiple items (set sizes 4, green, and 10, blue curves). Once the cue is presented, resources encoding uncued items are freed and the decaying sensory signal representing the target item is further integrated into VWM, still subject to limited total VWM resources but now without competition from other items. Due to exponential decay of the sensory signal, the increase in fidelity thus accrued changes rapidly with retention interval over the first few hundred milliseconds. At longer delays, the cue identifies the target only after the sensory signal has effectively disappeared, so the VWM signal representing the target item remains at the normalized level reflecting equal distribution between all items in the memory array, and memory dynamics consist only of the more gradual deterioration of fidelity due to accumulated noise in the encoded value.

Finally, the DyNR model predicts the presence of a set size effect on fidelity throughout the entire memory period, including the no delay (0 ms) condition in which the cue onset was coincident with stimulus offset, but not in the simultaneous cue condition. In the model, this behavior emerges as a consequence of two independent processes. First, at the end of stimulus presentation, items within smaller (lower set size) arrays are encoded in VWM with higher signal amplitude, reflecting normalization. This signal strength represents a baseline that can be supplemented by further integration of the sensory signal after an early cue. However, if the sensory decay is sufficiently rapid, then even if the cue is presented immediately the target representation will not attain the maximum amplitude (equivalent to set size of 1) starting from a lower baseline. Second, as described by Hick's law (*Hick, 1952*), it takes longer to identify the target item based on the cue as the number of alternatives increases (see *Alternative models* below for a formal test of this assumption). As a result, for higher set sizes, less sensory signal encoding the target item remains to be integrated into VWM once it has been identified.

## Model variants

We next focused on alternative explanations for the temporal dynamics observed in Experiment 1. Specifically, we examined whether the observed dynamics could be accounted for either solely by post-stimulus changes in neural signal amplitude or solely by noise-driven diffusion of stored values. To pre-empt our conclusions, we demonstrate that both components are needed to explain the observed dynamics in memory fidelity. Moreover, to more closely examine the role of diffusion in WM dynamics, we fit our neural model to an additional dataset collected in our lab (*Tomić et al., 2024*; see Appendix 4 for full details). This experiment used longer delays compared to those used in Experiment 1, and therefore precluded any beneficial effect of post-stimulus sensory information, while at the same time allowing the diffusion to operate over a longer period. This experiment allowed us to test whether diffusion is sufficient to account for human recall errors with longer memory delays.

### Fixed neural signal

A recent computational study on forgetting in VWM proposed that diffusion is sufficient to explain memory dynamics over delay (*Panichello et al., 2019*). To test for this, we developed two reduced versions of the DyNR model in which the diffusion process was solely responsible for memory fidelity dynamics. In both variants, the sensory signal terminated abruptly with stimuli offset, so the VWM signal encoding the stimuli was independent of the delay duration and equal to the limit imposed by normalization ($\hat{\gamma}_{wm}/N$). In the first variant, the diffusion rate was constant across set sizes, as in the full model. The formal model comparison demonstrated that the full DyNR model performed better than this simplified alternative (ΔAIC = 609.5).

In the second variant, we allowed the diffusion rate to increase proportionally with set size (for a similar proposal, see *Koyluoglu et al., 2017*). This model was again outperformed by the full DyNR model (ΔAIC = 666.4). Critically, both models tested here failed to qualitatively reproduce the observed nonlinear pattern of changes in recall error with time, notably overestimating recall error at the shortest delays by assuming no modulation in the representational signal (*Appendix 3—figure 1*).

## Diffusion

We developed two variants of the proposed neural model to test the role of diffusion. In the first variant, we completely omitted the diffusion process from the model to test whether the sensory signal modulation during the retention period is sufficient to explain temporal dynamics in recall fidelity. It could be argued that diffusion accounts for only minor changes in precision over brief delays as used here and, therefore, adds unnecessary complexity to the proposed model without improving the fit substantially. However, the formal model comparison revealed that the full DyNR model provides a better fit to human recall error compared to the matching model without diffusion (ΔAIC = 17.9).

The second variant was identical to the proposed model, except that we replaced the constant diffusion rate with a set-size-scaled diffusion rate by multiplying the right side of *Equation 6* by $N$. The model comparison showed that the full DyNR model also outperformed this variant (ΔAIC = 29.8). While both model variants qualitatively reproduced the increase in memory error with delay and set size, the pattern of variability was better explained by the model with a constant diffusion rate across set sizes. Although a more substantial diffusion effect could become apparent with longer delays than those used here, previous work demonstrated that noise-driven diffusion causes representations to deteriorate throughout the entire retention period (*Bouchacourt and Buschman, 2019*).

Finally, we examined the role of diffusion with longer memory intervals in a separate experiment using variable set sizes and memory intervals (1 s and 7 s) (for full details, see Additional dataset 1 in Appendix 4). We demonstrated that, once sensory information decayed completely, an accumulation of error during retention interval accounted for continuing memory deterioration. Together, the results presented here corroborate findings on the role of diffusion in temporal dynamics of recall fidelity (*Schneegans and Bays, 2018*).

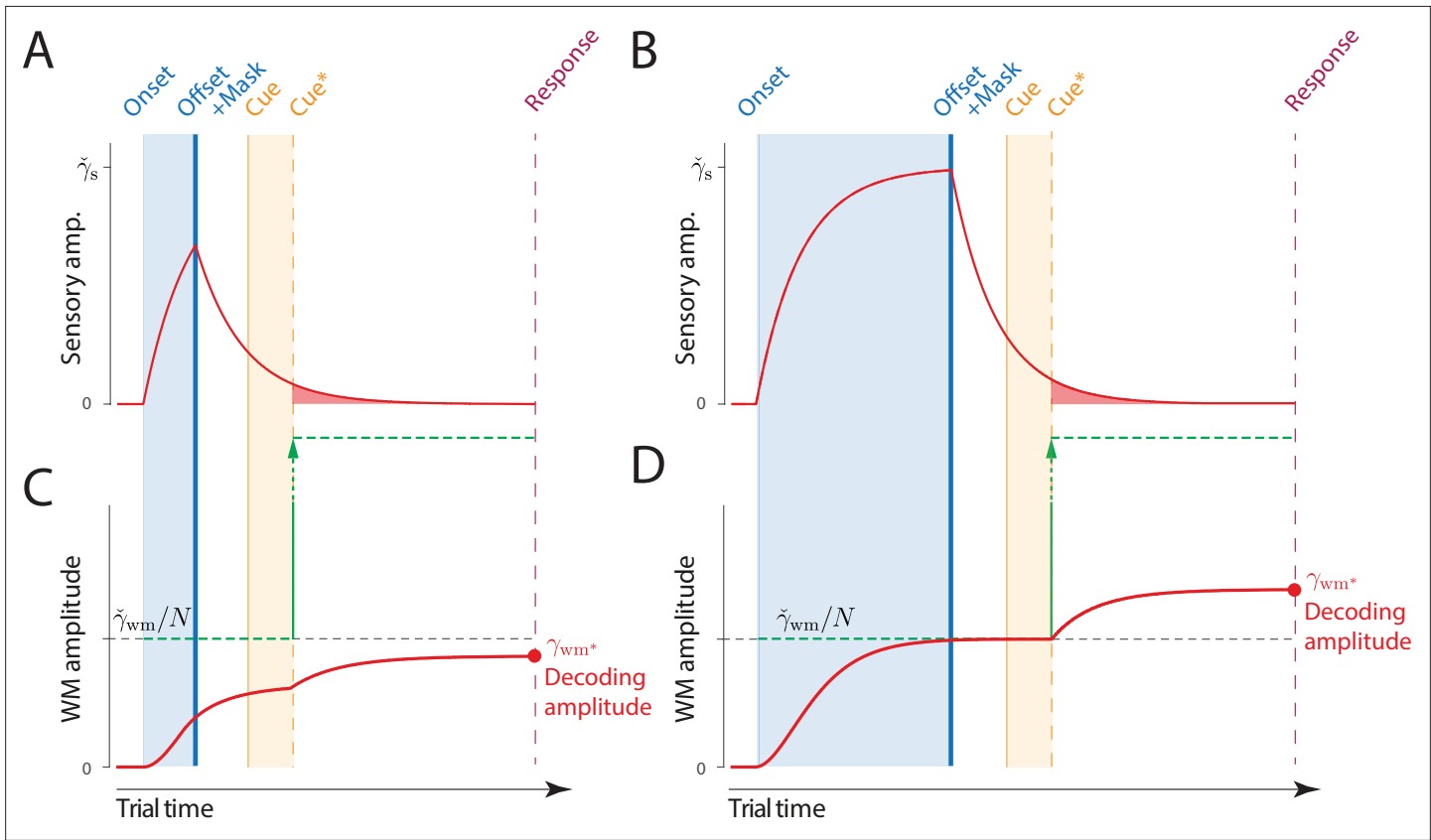

**Figure 5.** Time course of sensory and working memory (WM) gain with variable exposure duration. (**A, B**) The signal amplitude in the sensory population increases from stimulus onset, exponentially approaching the maximum sensory activity ($\check{\gamma}_s$). For shorter presentation durations (**A**) the attained amplitude at stimulus offset is only a fraction of the maximum (compare B, late offset). Following offset, sensory areas produce a decaying neural response, that is curtailed (faster decay) but not abolished by a backward mask. (**C, D**) Information about the stimulus is accumulated in WM from sensory activity. A shorter presentation (**C**) provides less sensory evidence for the initial accumulation of all items into visual working memory (VWM) (compare D, late offset), and subsequently less decaying sensory activity that can supplement VWM activity for the target item following the cue.

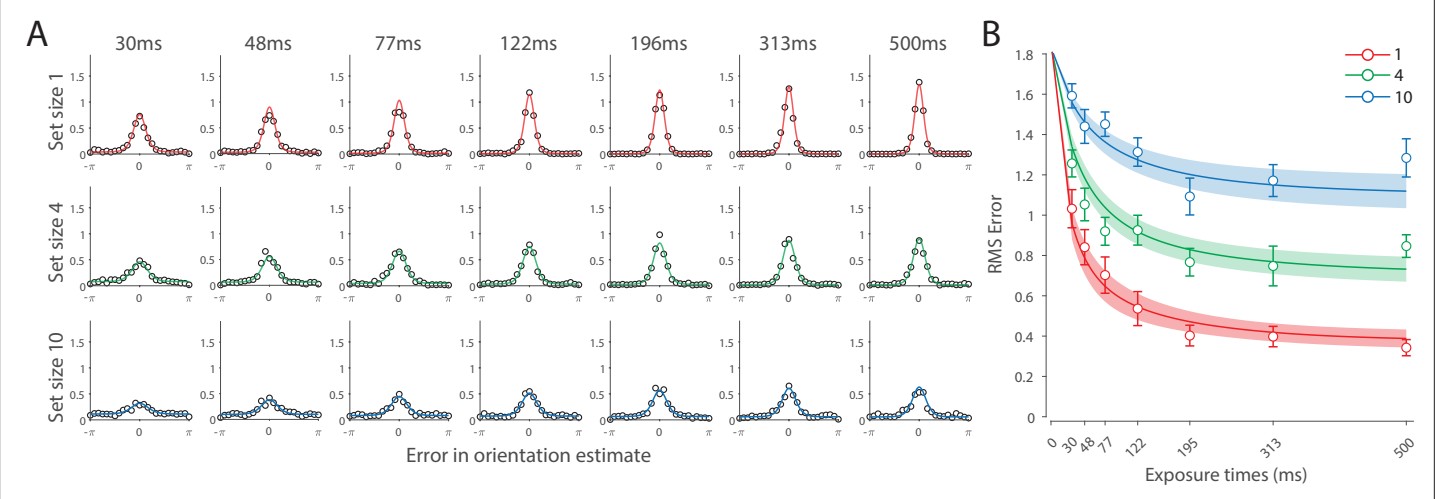

**Figure 6.** Experiment 2 results and modeling data show the consequences of varying set size and stimulus exposure time on visual working memory (VWM) reproduction error. (**A**) Empirical recall error distributions (black circles) and the dynamic neural resource (DyNR) model fits (colored curves). Different panels correspond to different set sizes (rows) and exposure durations (columns). (**B**) Corresponding RMS errors from experimental data (circles and error bars) and the DyNR model fits (curves and error patches). Error bars and patches indicate ±1 SEM. N = 13.

## Experiment 2: Exposure duration

In Experiment 2, we evaluated the encoding phase of VWM, by testing recall of orientation stimuli displayed in arrays of variable size presented for variable durations. In the DyNR model, increasing the sensory evidence by prolonging stimulus presentation has a favorable effect on later recall of stimulus, as more of that evidence can be accumulated into VWM. Importantly, this accumulation is also capped by the VWM resources available to store it (*Figure 5*).

### Experimental data

*Figure 6* shows the response error for different presentation durations and set sizes. Consistent with previous findings, response error can be seen to decrease with prolonged presentation duration, but increase as the number of items in memory increases. This was confirmed with a significant effect of display duration ($F_{(6,72)} = 29.01, p < 0.001, \eta^2 = 0.21$), set size ($F_{(2,24)} = 112.51, p < 0.001, \eta^2 = 0.54$), and their interaction ($F_{(12,144)} = 2.58, p = 0.004, \eta^2 = 0.019$). We further explored this interaction by first confirming that response error decreased with display duration within each set size ($F_{(6,72)} \geq 10.24, p < 0.001, \eta^2 \geq 0.26$). A consistent pattern was observed across set sizes, comprising an initial rapid decrease in response error over the briefest presentation times (first 200 ms), followed by saturation at prolonged exposure durations. Next, we calculated the change in recall error between the longest and the shortest display exposure within each set size, revealing that response error decreased more rapidly with display time as the number of items in memory decreased (ANOVA: $F_{(2,24)} = 7.79, p = 0.002, \eta^2 = 0.21$; corrected pairwise comparisons: $t_{1-4} = 3.65, p = 0.016, d = 0.87, t_{4-10} = 0.96, p = 0.72, d = 0.27$).

These results reveal the time course of information accumulation into VWM and forming of stable representations. We again identified several key characteristics of the dynamics of recall fidelity in the data (*Figure 6B*) to test against the DyNR model. Consistent with previous studies, we found recall fidelity changed with both presentation duration and the number of presented stimuli (*Bays et al., 2011*; *Shibuya and Bundesen, 1988*; *Vogel et al., 2006*). Specifically, as display duration increased from the shortest exposure, recall error showed an initial rapid decrease followed by a gradual leveling-off. As set size increased, the initial slope became shallower and the plateau occurred at a higher level of error.

### DyNR model

Curves in *Figure 6A and B* shows fits of the model with ML parameters (mean ± SE: population gain $\gamma$ = 188.5 ± 109.6, tuning width $\kappa$ = 10.2 ± 6.08, sensory rise time constant $\tau_{\text{rise}}$ = 0.33 ± 0.18, sensory

decay time constant $\tau_{\text{decay}}$ = 0.61 ± 0.19, VWM accumulation time constant $\tau_{\text{WM}}$ = 0.8 ± 0.34, cue processing constant $b$ = 0.2 s ± 0.09 s, base diffusion $\sigma_{\text{diff}}^2$ = 0.28 ± 0.08, spatial uncertainty time constant $\tau_{\text{spatial}}$ = 0.013 ± 0.004, swap probability p = 0.053 ± 0.01). The model provided an excellent quantitative fit to response distributions (*Figure 6A*) and RMSE (*Figure 6B*), successfully reproducing the pattern of changes with set size and presentation duration.

The model predicted that information from a visible stimulus accrues at a high rate immediately after the stimulus onset, consistent with observed changes in human recall error over stimulus durations up to 200 ms (*Figure 6*). This initial high encoding rate emerges naturally in the model due to the joint dynamics of sensory and VWM populations. In the sensory population, a low-pass temporal filter serves as a neural gain control mechanism, attenuating neural response to transient changes in stimuli (*Hess and Snowden, 1992*; *Hawken et al., 1996*). As a consequence, the neural response to stimulus onset increases exponentially (*Figure 5*). The information from sensory areas is accumulated into VWM, such that the accumulation rate is directly proportional to the difference between the current and saturating state (i.e. the rate is faster when accumulated information is far from the saturating state). Therefore, dynamics in the sensory and VWM population jointly account for the initial high rate of information extraction from stimuli, and its dependence on set size.

After the initial steep change, the model predicts that recall fidelity will asymptote. This was again observed in human behavior (*Figure 6*). Extending stimulus presentation beyond 200 ms had negligible impact on recall precision, consistent with previous studies (*Bays et al., 2011*). The model explains this behavior by describing how sensory signal and VWM accumulation independently saturate with time (*Figure 5*). Since the temporal filtering in the sensory population attenuates only high-frequency stimuli (i.e. very short presentations), with sufficient exposure, the sensory signal plateaus, resulting in a stable feedforward input to VWM. Similarly, VWM signal strength is subject to limits determined by normalization. Once the accumulated information reaches the normalized maximum set by the number of objects in memory, further accumulation of sensory evidence is not possible. Following the cue, a portion of the resource is freed, allowing the target representation to be further strengthened. However, because the sensory signal plateaus at longer exposures, the information available for integration after the cue remains constant across the longer exposures, supplementing normalized VWM signal by the same amount. The result is a plateau in fidelity that varies with set size.

## Model variants

We investigated whether post-stimulus sensory persistence contributed to the model fits in Experiment 2. We assumed that the signal persisting after stimulus offset would be impaired but not eliminated by the subsequent presentation of a noise mask in this experiment (*Kovács et al., 1995*). An alternative account suggests that the mask immediately terminates any stimulus-related signal. To test for this, we fit a variant of the DyNR model in which the sensory signal was terminated by the onset of the mask, providing a feedforward signal to VWM only for the period of the stimulus presentation. We found that the proposed DyNR model, in which some sensory signal persists after the mask onset, gave a better account of the data than this model variant (ΔAIC = 446.67). Although the alternative model captured the general pattern of changes in memory fidelity with exposure duration, it mispredicted fidelity at shorter exposures, in particular the effect of set size (*Appendix 3—figure 2A*).

A testable prediction of this alternative model is that the memory fidelity at recall should obey the neural normalization principle because there was no additional signal to supplement the presentation after initial encoding. To test for this, we additionally fitted each exposure condition separately using the original neural resource model with only three parameters (i.e. neural gain, tuning width, and swap probability). This model failed to predict actual fidelity levels at recall (*Appendix 3—figure 2B*), corroborating the findings of the model comparison.

Finally, to investigate the role of the post-stimulus sensory persistence on encoding dynamics, we fit the DyNR model to an additional dataset from *Bays et al., 2011* (for full details, see Appendix 5). This experiment aimed to investigate VWM dynamics during encoding, like our Experiment 2. In contrast to our Experiment 2, *Bays et al., 2011*, used a much longer delay interval (1100 ms vs 100 ms), precluding the possibility of further accumulation of sensory evidence following the cue. We expected that the DyNR model could account for memory dynamics in this study without any post-stimulus sensory activity. This was confirmed by accurately reproducing memory dynamics with

a model in which encoding into VWM relied only on sensory evidence during stimulus presentation (detailed results in Appendix 5).

## Alternative accounts

Having demonstrated the need for both post-stimulus sensory persistence and diffusion to account for empirical data, we next considered alternatives to our account of VWM accumulation and information read-out.

### Direct read-out of sensory information

In the DyNR model, recall fidelity is enhanced following the cue by integrating remaining sensory activity into capacity-limited VWM. As a consequence, response precision is bounded from above by the memory limit irrespective of the available sensory signal. An alternative possibility is that the decaying sensory representation can be directly read out following the cue to inform a response, bypassing WM limitations. To formalize this alternative model, we assumed that independent sensory and VWM representations would be optimally combined via summation of neural activity to yield population gain

$$\gamma^*_{\mathrm{sum}} = \gamma_{\mathrm{wm}}(t_{\mathrm{cue}*}) + \gamma_{\mathrm{s}}(t_{\mathrm{cue}*}) \tag{14}$$

The model is otherwise identical to the proposed DyNR model. A distinctive prediction of this model is that the precision of recall changes exponentially with delay for every set size, including 1 item (*Appendix 3—figure 3*). This prediction is qualitatively inconsistent with the pattern of results observed in Experiment 1, in contrast with the DyNR model which does not predict any beneficial effect of earlier cues with set size 1. This alternative model provided a worse fit to data from Experiment 1 ($\Delta$AIC = 164) and Experiment 2 ($\Delta$AIC = 84.6), for combined evidence favoring the DyNR model of $\Delta$AIC = 248.6.

### Cue processing

In the DyNR model, we assumed that identifying the target stimulus based on the cue is time-consuming, and becomes more so as the number of alternatives increases. Cue processing time encompasses perceptual, attentional, and decision components needed to interpret and act on the cue. We tested the necessity of this component by fitting a model variant in which VWM started accumulating evidence about the cued item at the moment of cue presentation. This model provided a worse fit to empirical data from both Experiment 1 ($\Delta$AIC = 84.5) and Experiment 2 ($\Delta$AIC = 107.5), for total evidence in favor of the DyNR model of $\Delta$AIC = 192 (*Appendix 3—figure 4*). We fit another variant in which cue processing time was constant across set sizes. This alternative provided a worse fit to the data in Experiment 1 ($\Delta$AIC = 191.6) and Experiment 2 ($\Delta$AIC = 105), for combined evidence $\Delta$AIC = 296.6 in favor of the full DyNR model that assumes cue processing time increases with set size. These results corroborate previous findings on the important role of cue processing time in models of attention (*Shih and Sperling, 2002*) and IM (*Sperling, 2018*).

### Constant accumulation rate

In the DyNR model, the rate of accumulation into VWM is proportional to the difference between the present VWM amplitude and the maximum normalized amplitude (*Equation 2*). An arguably simpler assumption is that the neural signal approaches saturation at a constant rate (*Boerlin and Denève, 2011*; *Beck et al., 2008*). In particular, the rate at which the signal representing an item is transferred to VWM is constant and depends only on the number of encoded items, i.e.,

$$\dot{\gamma}_{\mathrm{wm}}(t) = \begin{cases} \gamma_{\mathrm{s}}(t)/(M(t)\tau_{\mathrm{wm}}) & \text{if} \qquad \gamma_{\mathrm{wm}}(t) < \tilde{\gamma}_{\mathrm{wm}}/M(t) \\ 0 & \text{otherwise.} \end{cases} \tag{15}$$

The dependence on $M(t)$ satisfies the constraint that the neural resources in VWM are allocated at a constant rate, irrespective of the number of items. We applied this model to psychophysical data from both experiments (*Appendix 3—figure 5*) and found it provides a worse fit to the data from

Experiment 1 (ΔAIC = 11.5) and Experiment 2 (ΔAIC = 36.2), for combined evidence favoring the DyNR model with exponential saturation (ΔAIC = 47.7).

## Discussion

In the present study, we investigated the temporal dynamics of short-term recall fidelity. We conducted two new human psychophysical experiments and analyzed two existing datasets in order to characterize how recall errors are influenced by set size, stimulus duration, and retention interval. We developed a DyNR model to provide a mechanistic explanation of the observed behavior, capturing not only changes in overall fidelity but also the distribution of errors in the stimulus space and frequencies of swaps (intrusion errors). A key finding is that the benefit to recall precision observed at very short delays is due to additional post-cue integration of sensory information into WM, and that direct retrieval from sensory memory is unable to account for the empirical patterns of error.

### Sensory and WM dynamics during delay

In the first experiment we investigated the effects of brief unfilled delays on recall fidelity. With multi-item arrays, we observed that memory performance deteriorates precipitously over the first few hundred milliseconds after stimuli disappear, followed by a gradual leveling-off of error with longer delays (*Figure 4*). These results are consistent with previously reported patterns of memory dynamics (*Di Lollo and Dixon, 1988*; *Sperling, 1960*; *Bradley and Pearson, 2012*; *Neisser, 1967*), and estimates of sensory decay ranging between 100 ms and 400 ms (*Loftus et al., 1992*; *Lu et al., 2005*). Here, we shed new light on these results by taking a computational approach in explaining observed temporal dynamics, and asking what this superior recall's neural origin is and its relation with VWM. To answer these questions, we adapted the Neural Resource model of *Bays, 2014*, with a temporal component. The new DyNR model considers dynamics in a sensory neural population registering the stimuli and in a VWM population that stores the stimuli for later recall. Critically, our model assumes that objects encoded with limited precision into VWM can be flexibly supplemented with sensory activity following a recall cue, within a brief temporal window while the sensory population provides a feedforward input post-stimulus. The boost in the representational VWM signal predicts a behavioral benefit of early cues that is consistent with our data and a large corpus of previous experiments (*Coltheart, 1980*).

A common assumption in studies of visual short-term memory is that recall over brief delays is exclusively supported by one of two memory stores, IM or VWM (*Bradley and Pearson, 2012*; *Pratte, 2018*). In this account, a cue presented within the first few hundred milliseconds after stimulus offset allows observers to access high resolution but rapidly deteriorating representations in IM; once the information in IM has decayed, objects must be retrieved from the capacity-limited VWM store. Two pieces of evidence from the current study contradict this view and strongly suggest that recall depends on VWM from the moment objects disappear. First, the recall benefit of short delays was not observed for one item arrays. We propose that this behavior reflects the fact that, during encoding, the entirety of the VWM resource is allocated to a single object, leaving no free capacity for further enhancement based on the available sensory signal post-cue. Second, we found clear evidence that recall fidelity varied with set size even with no delay between stimulus offset and cue (0 ms condition). We argue that this arises from the set size dependence of representational fidelity in VWM, which is only incompletely compensated by integration of the decaying sensory signal post-cue, resulting in lower fidelity for higher set sizes. The DyNR model provides a successful quantitative account for these findings, which are in clear contrast with the traditional view of IM.

The rapid changes in fidelity over short delays can be distinguished from dynamics over longer retention intervals. A number of recent studies have observed a slow deterioration of VWM precision over the course of prolonged retention (*Schneegans and Bays, 2018*; *Pertzov et al., 2017*; *Rademaker et al., 2018*; *Ricker et al., 2014*; *Shin et al., 2017*; *Zhang and Luck, 2009*). The causes of this deterioration are still contested, but growing evidence links this behavior to noise-driven diffusion. At a mechanistic level, diffusion is considered a fundamental property of continuous attractor networks of the kind commonly associated with models of WM (*Brody et al., 2003*; *Khona and Fiete, 2022*). In such networks, memorized features are represented as persistent activity 'bumps' in the network's representational feature space. Over a memory delay, the activity bump is sustained by balanced

excitatory and inhibitory connections, while stochasticity in neural activity causes shifts of the bump along the feature dimension, taking the form of a random walk. Although we did not model the network processes governing stability and diffusion within neural populations, our implementation is consistent with random (Brownian) perturbation, as assumed by attractor models (see also *Schneegans and Bays, 2018*).

Our theoretical account of memory dynamics during delay differs from several existing models of forgetting, which emphasize diffusion as the dominant source of error in short-term memory (e.g. *Panichello et al., 2019*; *Koyluoglu et al., 2017*). To solely account for the observed data in Experiment 1, diffusion would need to be strongest early in the retention period, followed by a much weaker diffusion with longer delays. However, it is unclear why the diffusion rate would change, and particularly slow down, with time. Assuming a constant neural signal encoding the stimulus, this would predict greater variability in neural activity initially compared to the later period after stimuli offset. This is inconsistent with electrophysiological data showing relatively stable levels of spiking variability throughout the memory delay period (*Khanna et al., 2019*; *Chang et al., 2012*; *Hussar and Pasternak, 2010*). The results observed here are consistent with the proposal that modulation of neural signal over short memory intervals accounts for an abrupt change in response fidelity, while diffusion accounts for a slower change that grows with time.

In the present study, a model assuming a constant diffusion rate, independent of the stored number of items, was preferred to one in which diffusion rate increases linearly with set size. This is consistent with results of *Shin et al., 2017*, who did not find a significant effect of set size on the rate of memory deterioration. In contrast to that, *Koyluoglu et al., 2017*, recently proposed that the rate of diffusion scales with set size. However, this study did not account for the presence of swap errors, which we found to increase with retention interval as well as set size. To draw strong conclusions about the dependence of diffusion on set size would require a future study to disentangle the different sources of error that could, in principle, increase with delay.

## Sensory and WM dynamics during encoding

Having investigated memory degradation during the retention interval, in Experiment 2 we focused on the dynamics arising from accumulation of information during stimulus presentation. Using new psychophysical data, we showed that encoding of information into VWM is contingent on both presentation duration and the number of memorized stimuli. The observed patterns of data indicate that VWM encoding of elementary stimuli is mostly completed within the first 200 ms of presentation even at the largest set sizes, with minimal benefit of longer exposures, extending previous work (*Bays et al., 2011*; *Shibuya and Bundesen, 1988*; *Vogel et al., 2006*). This fast encoding process may have an adaptive role: with a key function of VWM to store and accumulate information across saccadic eye movements, an efficient system should deploy its resources within the duration of a typical gaze fixation (*Aagten-Murphy and Bays, 2018*; *Rolfs and Schweitzer, 2022*).

Our aim was again to move beyond the description of the encoding dynamics and to provide a biologically plausible neurocomputational account of these dynamics. To achieve that, we applied the same VWM accumulation process that operates post-cue to the sensory information during stimulus presentation. Using previously published and newly collected data, we show that a model in which VWM accumulates dynamical sensory input up to a fidelity limit can successfully account for patterns of human recall errors with variable set size and stimulus presentation. An important result of our modeling is that the accumulated information in VWM increases with a rate proportional to unfilled capacity. In particular, the model with such exponential accumulation provided a better fit than a model assuming a constant encoding rate. This parallels previous observations that models based on exponential-like extraction of information successfully characterize attention (*Bundesen, 1990*; *Sperling and Weichselgartner, 1995*), WM encoding (*Bays et al., 2011*; *Smith and Ratcliff, 2009*), memory updating (*Oberauer and Kliegl, 2006*), and broader cognitive processes (*Usher and McClelland, 2001*; *McClelland, 1979*). We hypothesize that this pattern represents an approach to an equilibrium state of balanced excitation from the sensory input and lateral inhibition within the VWM population, which is the basis for capacity of the memory system.

In Experiment 2, the longest presentation duration shows an upward trend in error at set sizes 4 and 10. While this falls within the range of measurement error, it is also possible that this is a meaningful pattern arising from visual adaptation of the sensory signal, whereby neural populations reduce

their activity after prolonged stimulation. This would mean less residual sensory signal would be available after the cue to supplement VWM activity, predicting a decline in fidelity at higher set sizes. Visual adaptation has previously been successfully accounted for by a type of delayed normalization model in which the sensory signal undergoes a series of linear and nonlinear transformations (*Zhou et al., 2019*). Such a model could in future be incorporated into DyNR and validated against psychophysical and neural data.

Our computational account of VWM encoding dynamics differs from several existing modeling frameworks aiming to explain similar data. For example, the theory of visual attention (TVA; *Bundesen, 1990*) assumes that visual stimuli participate in a parallel exponential race toward limited VWM. Like the DyNR model, TVA assumes a form of normalization in the sense that the speed with which items race toward VWM depends on the number of items in the visual field. Unlike our dynamic model, TVA is not a theory of VWM, and it considers VWM only as a storage for categorizations of visual objects. In particular, TVA takes into account the limits of VWM but does not specify why or how these limitations arise. Finally, TVA considers whether an object was selected for entry into VWM in an all-or-none fashion; our dynamic model is mostly concerned with the fidelity of representations. A somewhat alternative account of VWM encoding is provided by the competitive interaction theory (CIT; *Sewell et al., 2014*), which is similarly based on the signal detection theory and principles of normalization (*Reynolds and Heeger, 2009*). Like TVA, CIT is mostly focused on item selection and merely incorporates a concept of VWM capacity derived from object-based models of VWM. Although CIT had success in accounting for behavioral data from a two-alternative orthogonal discrimination task using up to four items and a limited range of encoding times, it remains an open question whether this model can account for error distributions as measured in a continuous report task, and a larger range of set sizes and stimulus exposures. Importantly, compared to both TVA and CIT, the DyNR model is strongly rooted in and inspired by findings from neuroscience. This not only adds to the biological plausibility of our model but also allows future studies to test the model's predictions using physiological methods.

## Neural mechanisms

The theory presented here generalizes the Neural Resource model of *Bays, 2014*, a simple encoding-decoding model in which visual features are represented in the noisy spiking activity of neural populations (*Pouget et al., 2000*), and where the activity representing each feature scales inversely with the total number of representations, consistent with the prevalence of normalization mechanisms in the brain and observations from single-neuron recording (*Buschman et al., 2011*) and fMRI decoding (*Sprague et al., 2014*) studies. The population coding in the model is based on an abstract idealization of neural response functions. Nevertheless, it has recently been shown that more realistic population coding schemes that allow for heterogeneity in neural tuning curves and correlated spiking activity as observed in visual cortex maintain the key predictions of the idealized model (*Taylor and Bays, 2020*; *Schneegans et al., 2020*). This may be seen as a consequence of the different population codes inducing a common representational geometry (*Kriegeskorte and Wei, 2021*).

We adapted the stationary VWM model by first incorporating a sensory population that provides an input drive to the VWM population. In parallel with neurophysiological observations, a common approach is to model these dynamics with a low-pass filter which acts like a neural gain modulation mechanism (*Hawken et al., 1996*). As a consequence, the sensory response to stimulus onset and offset is an exponential rise and decay in activity, respectively. The decaying component of the response has been recognized as a neural substrate of visual persistence and IM (*van Kerkoerle et al., 2017*; *Teeuwen et al., 2021*). Here, we modeled sensory decay with an exponential function (*Zylberberg et al., 2009*), although other forms of decay have been proposed. For example, *Loftus et al., 1992*, showed that iconic decay could be better captured using a gamma survival function, a generalization of exponential decay that could simply be implemented in our neural model by replacing a single filter with a cascade of exponential low-pass filters.

In addition to the dynamics in the sensory population, two features of VWM introduce additional dynamics in representation fidelity: the accumulation of information (discussed above) and the diffusion of representations owing to accumulated noise. Although we did not aim to model the neural processes behind diffusion, our implementation is consistent with the consequences of neural variability in attractor networks (*Burak and Fiete, 2012*; *Khona and Fiete, 2022*). Converging neural

evidence demonstrating such diffusion has been observed using single-unit neural recording in monkeys (*Wimmer et al., 2014*), as well as EEG (*Wolff et al., 2020*) and fMRI (*Lim et al., 2019*; *Yu et al., 2020*) studies in humans.

As well as being implicated in higher cognitive processes including VWM (*Buschman et al., 2011*; *Sprague et al., 2014*), divisive normalization has been shown to be widespread in basic sensory processing (*Bonin et al., 2005*; *Busse et al., 2009*; *Ni and Maunsell, 2017*). The DyNR model presently incorporates the former but not the latter type of normalization. While the data observed in our experiments do not provide evidence for normalization of sensory signals (note comparable recall errors across set size in the simultaneous cue condition of Experiment 1), this may be because sensory suppressive effects are localized and our stimuli were relatively widely separated in the visual field: future research could explore the consequences of sensory normalization for recall from VWM using, e.g., center-surround stimuli (*Bloem et al., 2018*).

Following onset of a stimulus, the visual signal ascends through visual areas via a cascade of feedforward connections. This feedforward sweep conveys sensory information that persists during stimulus presentation and briefly after it disappears (*Lamme et al., 1998*). Simultaneously, reciprocal feedback connections carry higher-order information back toward antecedent cortical areas (*Lamme and Roelfsema, 2000*). In our psychophysical task, feedback connections likely play a critical role in orienting attention toward the cued item, facilitating the extraction of persisting sensory signals, and potentially signaling continuous information on the available resources for VWM encoding. While our computational study does not address the nature of these feedforward and feedback signals, a challenge for future research is to describe the relative contributions of these signals in mediating transmission of information between sensory memory and WM (*Semedo et al., 2022*).

Our model makes a clear distinction between dynamics in sensory and VWM populations, however, it remains agnostic as to whether the populations have the same or different anatomical locus (*Rademaker et al., 2019*). Albeit inspired by the properties of orientation-selective neurons in area V1, population tuning of this kind is a common coding motif across the brain (*Pouget et al., 2000*). While it could be considered efficient to use already specialized circuits to maintain as well as process visual information, it is still debated whether sensory areas are a feasible candidate for memory storage (*Serences, 2016*; *Xu, 2017*). While some studies have focused on prefrontal (*Goldman-Rakic, 1995*), parietal (*Bettencourt and Xu, 2016*), or occipital (*Harrison and Tong, 2009*) cortices as the primary locus of VWM, others argue for distributed storage by demonstrating that VWM contents can be decoded from imaging signals originating in multiple brain areas (*Christophel et al., 2018*).

## Representational dynamics of cue-dimension features

Memory retrieval failures in which a non-cued item is reported in place of the intended target represent an important source of error in VWM recall. These swap errors occur more often at higher set sizes and when spatial confusability is high (*Bays et al., 2009*; *Emrich and Ferber, 2012*; *Rerko et al., 2014*; *Bays, 2016b*), as predicted by models in which they arise from uncertainty in the recall of cue-dimension features leading to incorrect selection of an item in memory (*Schneegans and Bays, 2017*; *McMaster et al., 2022*). In the current study, we assumed memory for spatial location (the cue feature) undergoes similar dynamics to memory for orientation (the report feature), and in particular that spatial information degrades with retention time (*Schneegans and Bays, 2018*), leading to changes in swap error frequency with delay interval. Similarly, during encoding the fidelity of spatial representation increases with the accumulation of sensory evidence (*Zimmermann et al., 2013*), reducing the uncertainty at retrieval and consequently swap errors at longer stimulus exposure. Although we did not explicitly model the neural signals representing location, the modeled dynamics in the probability of swap errors were consistent with those of the primary memory feature. We provided a more detailed neural account of swap errors in our earlier works that is theoretically compatible with the DyNR model (*Schneegans and Bays, 2017*; *McMaster et al., 2022*).

The DyNR model successfully captured the observed pattern of swap frequencies (intrusion errors). The only notable discrepancy between DyNR and the three-component mixture model (*Appendix 2—figure 1*) arises with the largest set size and longest delay, although with considerable interindividual variability. As the variability in report dimension increases, the estimates of swap frequency become more variable due to the growing overlap between the probability distributions of swap and non-swap

responses. This may explain apparent deviations from the modeled swap frequencies with the highest set size and longest delay where orientation response variability was greatest.

## Removal of information from WM

In the DyNR model, taking advantage of early cues requires rapid removal of the VWM signal associated with uncued items, to admit further accumulation of activity encoding the cued item. To achieve this, an active process of selective content elimination may be required (*Oberauer, 2018*), as opposed to a passive decay of uncued representations during the post-cue interval. Mounting evidence for such active removal has been provided at the behavioral (*Williams et al., 2013*) and neural (*LaRocque et al., 2013*) level. Importantly, studies show that a functional role of such active removal is to release resources allocated to the uncued representations, facilitating the encoding of new information (*Taylor et al., 2023*; *Souza et al., 2014*). The fast reallocation of neural resources assumed by the DyNR model is consistent with such a description of active removal.

# Methods

## Participants

A total of 23 naive observers (12 females, 11 males; aged 18–34) took part in the study after giving informed consent in accordance with the Declaration of Helsinki. Ten observers participated in Experiment 1 and 13 observers participated in Experiment 2. Volunteers were recruited through the Cambridge Psychology research sign-up system. All observers reported normal color vision and normal or corrected-to-normal visual acuity, and were remunerated £10/hr for their participation. Procedures were approved by the University of Cambridge Psychology Research Ethics Committee (approval number PRE.2015.099).

## General methods

### Experimental setup

Stimuli were presented on a 69 cm gamma-corrected LCD monitor with a refresh rate of 144 Hz. Participants were seated in a dark room and viewed the monitor at a distance of 60 cm, with their head supported by a forehead and chin rest. Responses were collected using Magic Trackpad 2, a pointing device (16×11.5 cm$^2$) with a tactile sensor operating at ~90 Hz (Apple Inc). Eye position was monitored online at 1000 Hz using an infrared eye tracker (SR Research). Stimulus presentation and response registration were controlled by a script written in Psychtoolbox and run using Matlab (The Mathworks Inc).

### Stimuli

Memory stimuli consisted of randomly oriented Gabor patches (wavelength of the sinusoid, 0.65° of visual angle; s.d. of Gaussian envelope, 0.5°) presented on a uniform mid-gray background. The contrast of Gabor patches varied between experiments (see below). Memory stimulus positions were randomly chosen from a set of 10 equidistant locations on the perimeter of an invisible circle with radius 6° centered at fixation. At the start of each trial, a black fixation annulus was shown ($r = 0.15°$ and $R = 0.25°$) in the display center. Once steady fixation was registered, the size of the inner radius increased ($r = 0.2°$). Observers perceived this change as the annulus becoming thinner. The fixation annulus then stayed visible throughout the trial. Items were cued for recall by displaying a black arrow (2° length) extending from the center of the display and pointing to one of the previously occupied locations without overlapping with it.

### Procedure

Each trial started with presentation of the central fixation annulus. Observers were required to maintain gaze fixation for 500 ms within a radius of 2° around the central annulus in order for a trial to proceed. Following stable fixation, the appearance of the fixation annulus changed, indicating that the memory array would appear in 500 ms. The memory sample array consisting of 1, 4, or 10 randomly oriented Gabor patches was then presented. This was followed by a delay period and finally a cue

display, indicating to observers to report the memorized orientation of an item previously displayed at the indicated location.

Observers were instructed to reproduce the remembered orientation as accurately and as quickly as possible by executing a single movement of their index fingertip over the surface of the touchpad located centrally in front of them. Simultaneously with the observer's movement, a blue line appeared on the screen, extending from the center of the screen and mimicking the observer's response in real time. The response was terminated if one of the following conditions was satisfied: the observer stopped movement for 500 ms; the observer lifted their finger from the touchpad; or the response line reached the edge of the display. This was followed by a feedback display, consisting of the actual orientation (shown with a white line) and reported orientation (shown with a blue line) overlaid at the location of the cued item. The recalled orientation was calculated as the angle of the line connecting a starting point and an endpoint of hand movement on the touchpad.

Observers were required to maintain central fixation during the stimulus presentation and delay phase. If gaze position deviated by more than 2° a message appeared on the screen, and the trial was aborted and restarted with newly randomized orientations. Participants completed the task in blocks of 50 trials, and each block corresponded to one experimental condition. The order of blocks was randomized for every observer. At the beginning of the testing session observers familiarized themselves with the task and experimental setup by doing at most 50 practice trials.

## Experiment 1

In Experiment 1 we investigated the temporal dynamics of VWM fidelity over short delays by presenting observers with sets of stimuli of variable size and then cueing one of them for recall after a variable delay relative to the stimuli offset. A typical trial sequence is shown in *Figure 3A*. The memory sample array (Michelson contrast = 0.5) was presented for 200 ms. In 50% of trials, the stimuli changed phase (by 180°) and contrast (Michelson contrast = 1) for the last 50 ms of presentation, while remaining at the same orientation. This manipulation was intended to minimize retinal after-effects (see, e.g., *Kelly and Martinez-Uriegas, 1993*, for similar techniques and Appendix 1 for validation). The stimuli offset was followed by a variable blank delay of 0, 100, 200, 400, or 1000 ms, after which one item was cued for recall. In one additional condition, the cue was instead presented simultaneously with the memory sample array, indicating an item while it was still visible on the screen (*Figure 3B*).

Each observer completed a total of 1800 trials, split into 36 blocks. The experiment was organized such that half of the observers first completed 18 blocks with phase shift (see above), and the other half first completed blocks without phase shift. Except for this constraint, block order was randomized for every observer. The testing was divided into four equal testing sessions, each lasting approximately 1.5 hr, with a separation of at least 1 day between sessions.

## Experiment 2

In Experiment 2 we investigated the temporal dynamics of VWM fidelity during encoding. To this end, we displayed oriented stimuli for a variable duration and in sets of variable size. The experiment was similar to the previous experiment with a few exceptions (*Figure 3C*). Each trial started with a presentation of a fixation annulus, followed by a memory array (Michelson contrast = 0.3). The stimuli stayed on the screen for a variable duration of 30, 48, 77, 122, 196, 313, or 500 ms, and were then replaced by noise masks (100 ms). Mask stimuli consisted of white noise at full contrast, windowed with a Gaussian envelope (0.5° s.d.) and flickering at 35 Hz. At the offset of the masking stimuli, one memory item was cued for recall. Each observer completed 21 blocks, for a total of 1050 trials. Blocks were spread over two testing sessions, each lasting approximately 1.5 hr, and taking place on different days. Observers completed 10 blocks in the first, and the remaining 11 blocks in the second session.

## Acknowledgements

We thank George Sperling and Sebastian Schneegans for helpful discussion, Robert Taylor for help with Bayesian hierarchical modeling, and Jessica McMaster for help with data collection. We used resources provided by the Cambridge Service for Data Driven Discovery (CSD3) operated by the University of Cambridge Research Computing Service. This research was supported by the Wellcome Trust (grant 106926 to PMB).

# Additional information

## Funding

| Funder | Grant reference number | Author |
| --- | --- | --- |
| Wellcome Trust | 10.35802/106926 | Paul M Bays |

The funders had no role in study design, data collection and interpretation, or the decision to submit the work for publication. For the purpose of Open Access, the authors have applied a CC BY public copyright license to any Author Accepted Manuscript version arising from this submission.

## Author contributions

Ivan Tomić, Conceptualization, Data curation, Formal analysis, Investigation, Methodology, Software, Visualization, Writing – original draft, Writing – review and editing; Paul M Bays, Conceptualization, Formal analysis, Funding acquisition, Methodology, Project administration, Resources, Software, Supervision, Visualization, Writing – review and editing

## Author ORCIDs

Ivan Tomić ⓘ https://orcid.org/0000-0001-6046-9642
Paul M Bays ⓘ https://orcid.org/0000-0003-4684-4893

## Ethics

Human subjects: Procedures were approved by the University of Cambridge Psychology Research Ethics Committee (approval number PRE.2015.099). All subjects gave informed consent in accordance with the Declaration of Helsinki.

Reviewer #2 (Public Review): https://doi.org/10.7554/eLife.91034.3.sa1
Reviewer #3 (Public Review): https://doi.org/10.7554/eLife.91034.3.sa2
Author response https://doi.org/10.7554/eLife.91034.3.sa3

# Additional files

## Supplementary files

• MDAR checklist

## Data availability

Data and code related to this study will be made available at https://doi.org/10.17863/CAM.95223.

The following dataset was generated:

| Author(s) | Year | Dataset title | Dataset URL | Database and Identifier |
| --- | --- | --- | --- | --- |
| Tomić I, Bays PM | 2024 | Research data supporting 'A dynamic neural resource model bridges sensory and working memory' | https://doi.org/10.17863/CAM.95223 | Apollo - University of Cambridge Repository, 10.17863/CAM.95223 |

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

# Appendix 1

## Minimizing retinal after-effects

We assessed the method of minimizing retinal afterimages by repeating all measurements, with the exception of not using phase shift of stimuli (*Figure 3A*). We predicted retinal afterimages could serve as an additional source of information, but only for a brief period after stimuli offset. Therefore, here we expected to see better performance for brief delays compared to conditions with phase shift. *Appendix 1—figure 1A* shows recall error increased with both set size and delay. Both of these effects were statistically significant, as well as their interaction (set size: $F_{(2,18)} = 47.3, p < 0.001, \eta^2 = 0.31$; delay time: $F_{(5,45)} = 48.4, p < 0.001, \eta^2 = 0.26$; interaction: $F_{(10,90)} = 21.3, p < 0.001, \eta^2 = 0.14$), reminiscent of findings for data with phase shift.

Next, we focused on the comparison of conditions with and without phase shift of stimuli (*Appendix 1—figure 1B*). We illustrate the difference in performance by subtracting RMSE obtained in the condition without phase shift (*Figure 4B*) from RMSE shown in *Appendix 1—figure 1A*. Negative values indicate better performance in a condition without phase shift. As predicted, the overall pattern of data suggested performance was comparable for 1 item across all delays, and for all set sizes for extreme delays (simultaneous presentation and 1000 ms), indicated by the difference values around 0. We confirmed the difference in recall error for 1 item across all delays did not differ consistently with and without phase shift, as neither phase shift $(F_{(1,9)} = 0.03, p = 0.86, \eta^2 < 0.001, BF_{incl} = 0.143)$ nor the interaction of phase shift and delay $(F_{(5,45)} = 0.41, p = 0.89, \eta^2 = 0.00, BF_{incl} = 0.042)$ reached significance. Based on this result, we conducted all remaining analyses using only the remaining two set sizes. We ran separate repeated measures ANOVAs for each delay using phase shift and set size as factors. The pattern of results we observed was clear: performance was comparable with and without phase shift with the simultaneous presentation and 1000 ms delay (phase shift, $F_{(1,9)} \leq 1.08, p \geq 0.33, \eta^2 \leq 0.002, BF_{excl} \geq 3.62$; interaction, $F_{(2,18)} \leq 0.8, p \geq 0.44, \eta^2 \leq 0.02, BF_{excl} \geq 3.39$), while for the remaining intermediate delays recall error was consistently lower when phase shift was omitted (phase shift, $F_{(1,9)} \geq 5.8, p \leq 0.039, \eta^2 \geq 0.06$; interaction, $F_{(1,9)} \leq 2.8, p \geq 0.13, \eta^2 \leq 0.001$).

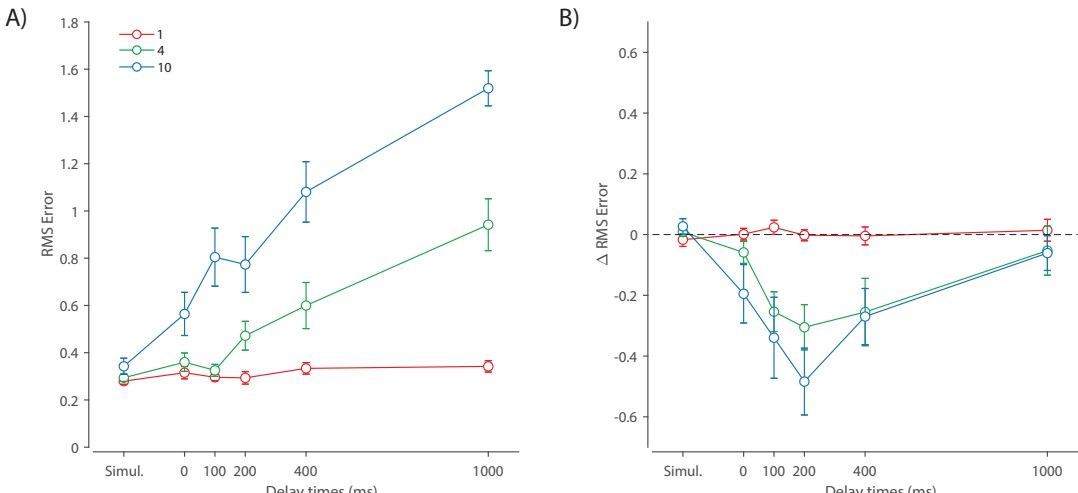

**Appendix 1—figure 1.** Minimizing retinal after-effects. Cancelling retinal afterimages. (**A**) Experiment 1 RMSE for trials without phase shift. (**B**) Differences in RMSE between trials with and without phase shift across set size and delay conditions. Negative values indicate better performance in the condition without phase shift. Error bars indicate ±1 SEM. N = 10.

Taken together, performance with and without phase shift of stimuli was comparable in perceptual condition (simultaneous presentation) and with the longest delay, suggesting phase shift did not change visibility or encoding of information into VWM. In contrast, we found strong evidence that observers had access to an additional source of information over intermediate delays when phase shift was not used, demonstrated by a better recall performance from 0 ms to 400 ms delay. Specifically, this source of information was available immediately after stimuli offset and was

short-lived, consistent with the theoretical description of retinal afterimages (*Tsuchiya and Koch, 2005*).

## Appendix 2

## Swap error estimates

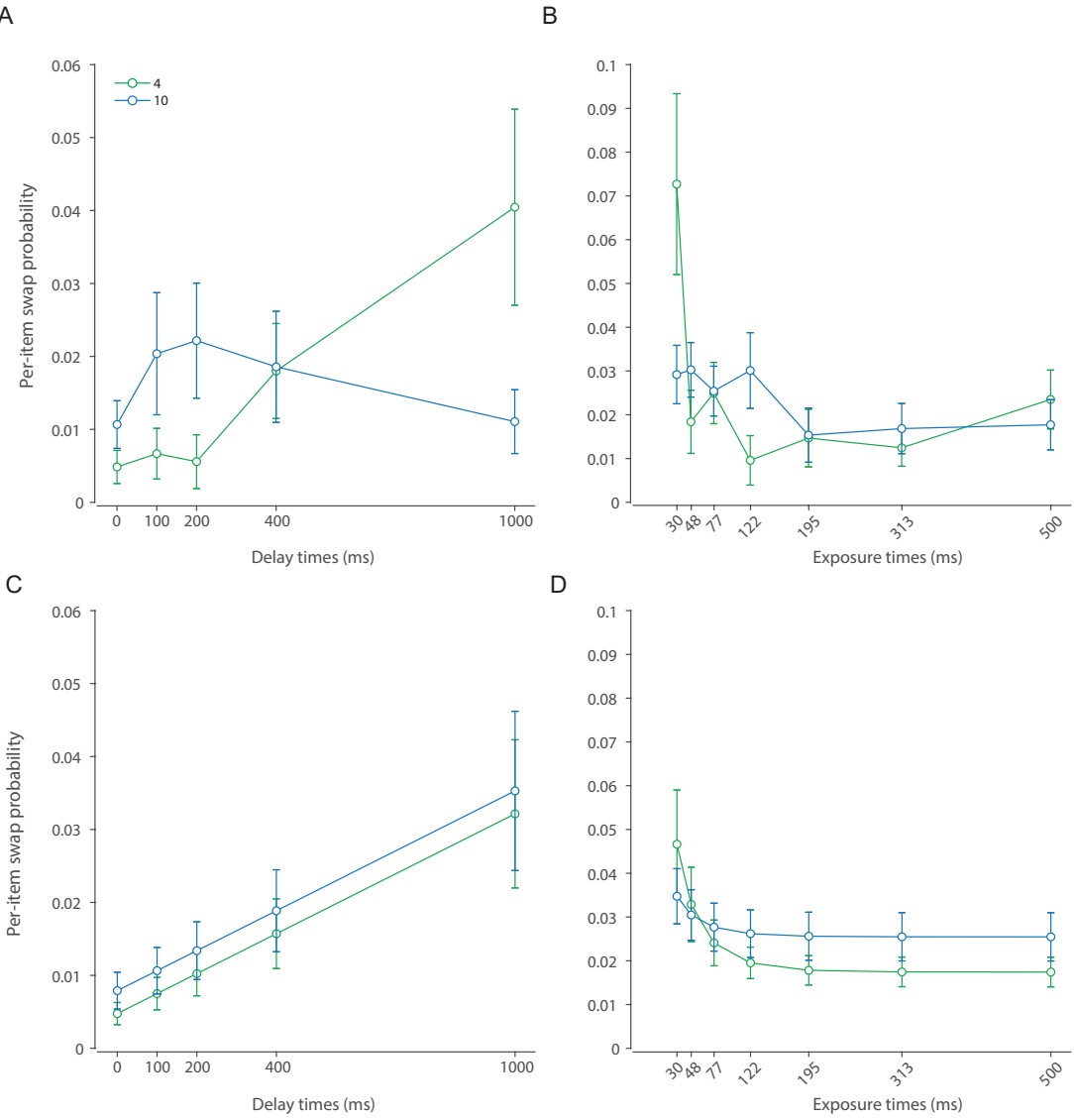

**Appendix 2—figure 1.** Swap error estimates. (**A and B**) Probability of swap errors estimated from empirical data using the three-component mixture model (***Bays et al., 2009***) in Experiment 1 (**A**) and Experiment 2 (**B**). (**C and D**) Probability of swap errors in best-fitting dynamic neural resource (DyNR) model in Experiment 1 (N = 10) (**C**) and Experiment 2 (N = 13) (**D**). Error bars indicate ±1 SEM.

## Appendix 3

## Alternative models' fits

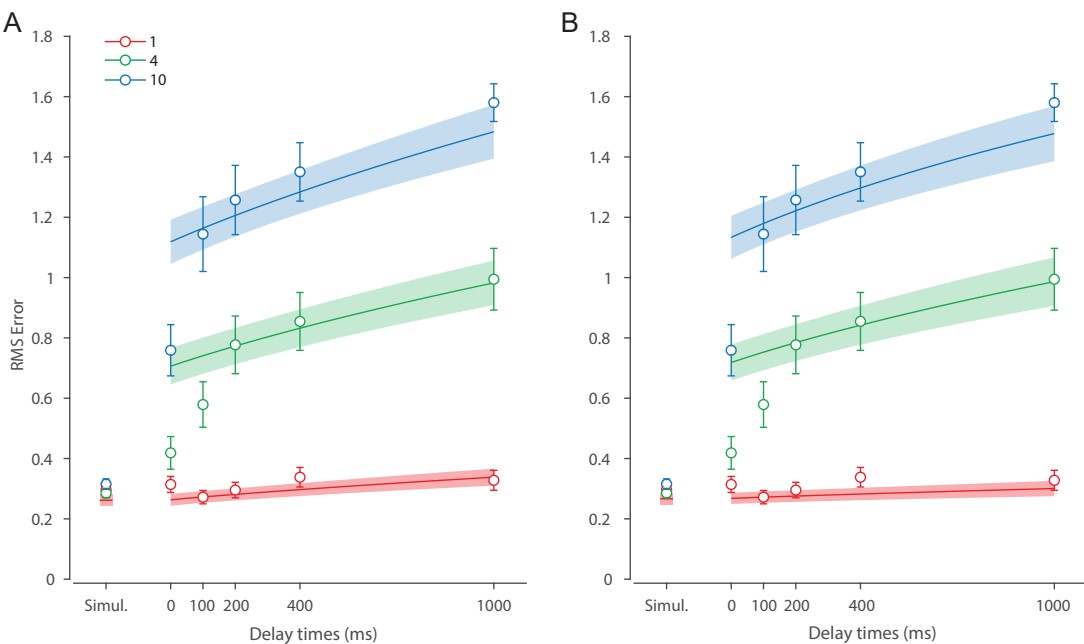

**Appendix 3—figure 1.** Experiment 1 behavioral data and model fit for the dynamic neural resource (DyNR) model without sensory persistence after stimulus offset. (**A**) A version of the DyNR model with equal diffusion across set sizes. (**B**) A version of the DyNR model with diffusion that scales with set size. Error bars and patches indicate ±1 SEM. N = 10.

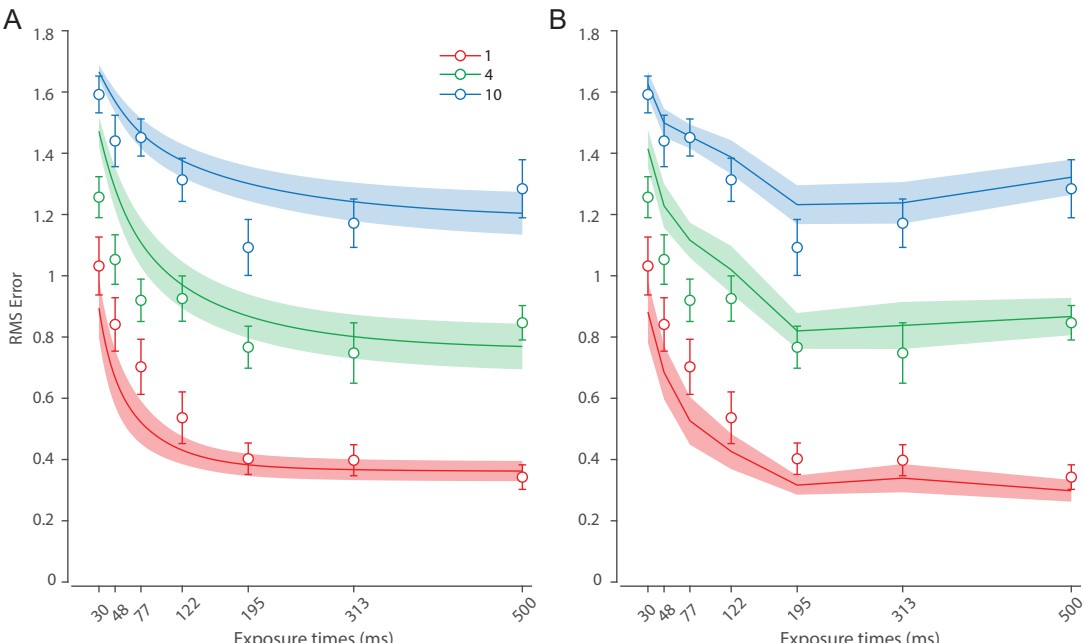

**Appendix 3—figure 2.** Experiment 2 behavioral data and model fit for the neural model without sensory persistence after stimulus offset. (**A**) A version of the dynamic neural resource (DyNR) model without sensory persistence. (**B**) Separate fits of the simplified neural model to each exposure time. Error bars and patches indicate ±1 SEM. N = 13.

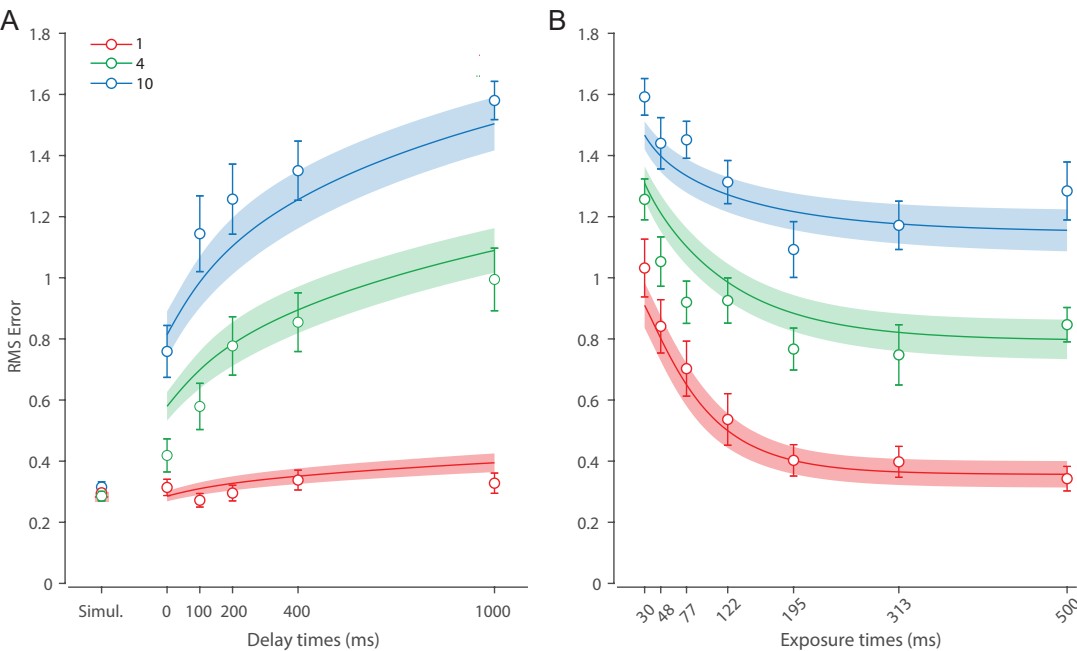

**Appendix 3—figure 3.** Behavioral data and model fit for a neural model with the direct read-out of information from sensory memory for (**A**) Experiment 1 (N = 10) and (**B**) Experiment 2 (N = 13). Error bars and patches indicate ±1 SEM.

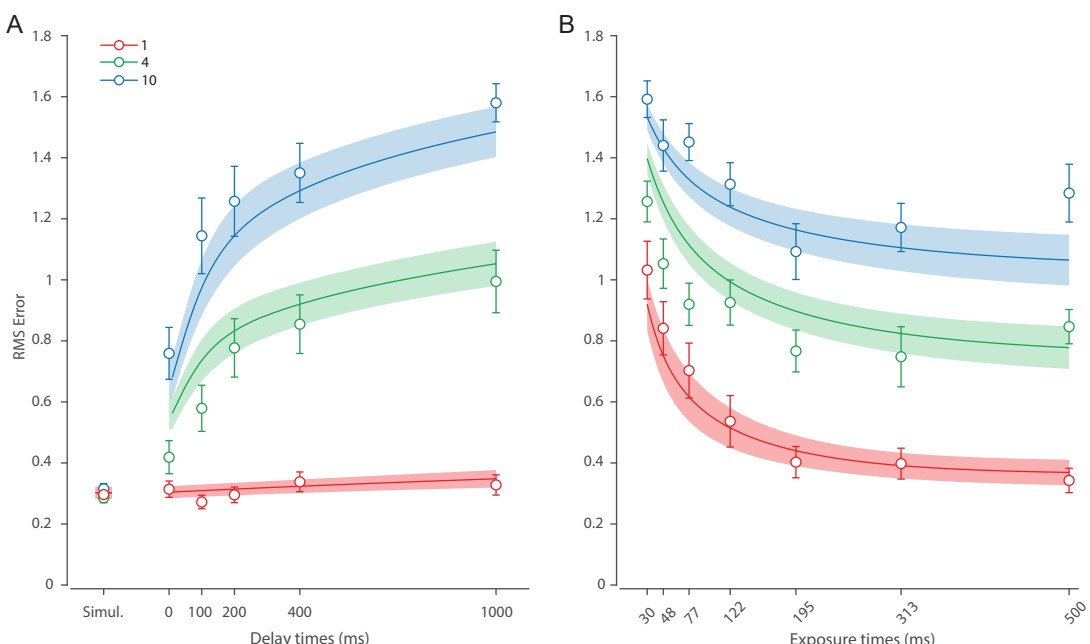

**Appendix 3—figure 4.** Behavioral data and model fit for the dynamic neural resource (DyNR) model without the cue processing time for (**A**) Experiment 1 (N = 10) and (**B**) Experiment 2 (N = 13). Error bars and patches indicate ±1 SEM.

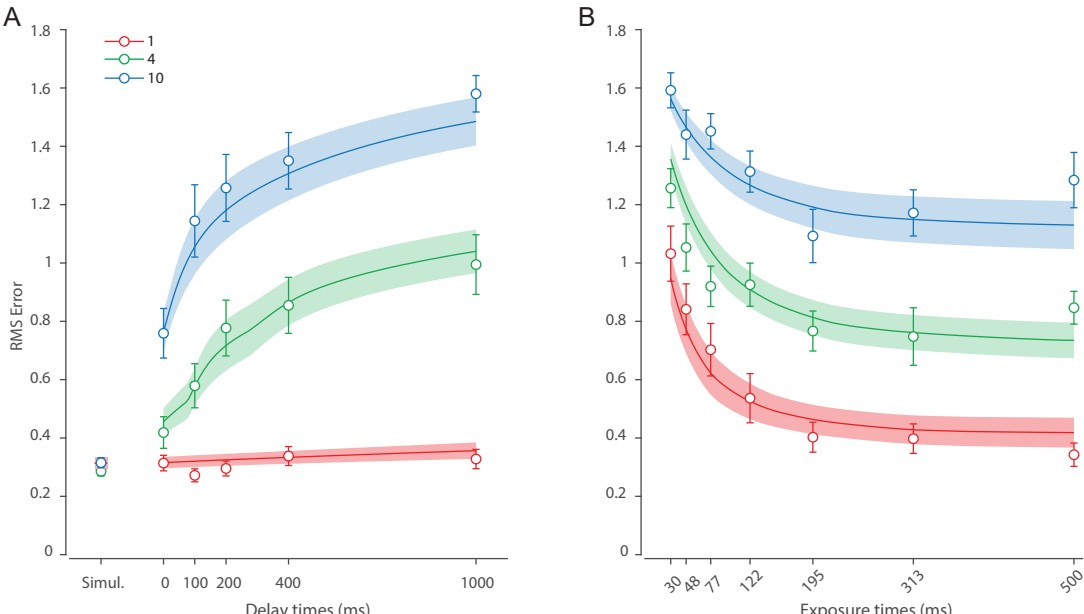

**Appendix 3—figure 5.** Behavioral data and model fit for a neural model with constant accumulation of information into working memory (WM) for (**A**) Experiment 1 (N = 10) and (**B**) Experiment 2 (N = 13). Error bars and patches indicate ±1 SEM.

# Appendix 4

## Additional dataset 1

To further investigate the role of diffusion in memory dynamics, we analyzed an additional dataset collected in our lab (*Tomić et al., 2024*). In this experiment we varied the set size and delay duration similar to Experiment 1. In contrast to Experiment 1, we used longer memory delays, which allowed us to examine the diffusion mechanism on a more suitable time scale. Moreover, memory delays used in this study are out of reach of the decaying sensory information, enabling us to investigate the diffusion without changes in the neural signal strength post-cue.

## Methods

Ten observers (six females, four males, aged 18–34) took part in this experiment. The data for this experiment was collected using the same equipment and the testing setting as described for the main experiments. A typical trial sequence is illustrated in *Appendix 4—figure 1*. Each trial began with the presentation of a central annulus which served as a fixation point. Once a stable fixation was achieved, the inner annulus radius changed indicating that stimuli would appear in 500 ms. The memory sample array was then presented for a duration of 500 ms. The array consisted of one or three randomly oriented black bars (length 2.8°). Each bar was positioned in one of six predetermined locations equally distributed around the circle with a radius of 5° around center of the screen. Each bar was presented along with a placeholder circle (radius 1.5°).

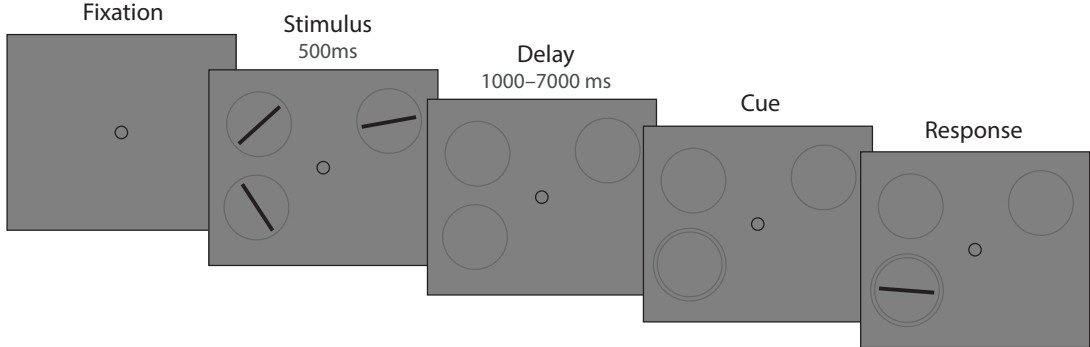

**Appendix 4—figure 1.** Experimental procedure. Stimuli are not drawn to scale.

Memory array presentation was followed by a memory delay during which fixation circle and placeholders stayed visible. The retention interval was either 1 s or 7 s long. After that, one stimulus was randomly cued for recall. The cue consisted of a second, larger circle drawn around one of the placeholders. Observers were instructed to start rotating a response dial (Griffin Technology PowerMate USB) once they were ready to respond. After the rotation of the response dial was detected, a randomly oriented black bar was displayed within the placeholder. Observers were instructed to rotate the dial until the displayed bar matched the remembered orientation of the cued item. Observers confirmed their response by pressing the dial. Trials with different set sizes and delay durations were randomly interleaved.

Eye movements were monitored from the beginning of the trial until stimuli offset, and observers were required to hold steady fixation during that period. If the gaze position deviated by more than 2° a message appeared on the screen and the trial was aborted and restarted with new orientations. Each observer completed 700 trials, divided into two sessions and each consisting of seven equal blocks. Two sessions were separated by at least 1 day, and each lasted approximately 1 hr. At the beginning of each session observers familiarized themselves with the task and experimental setup by doing at most 50 practice trials.

## Results

### Behavioral data

Recall performance is shown in *Appendix 4—figure 2*. As predicted, response error increased with set size and memory delay. A repeated measures ANOVA revealed a significant effect of set size ($F_{(1,9)} = 111.17, p < 0.001, \eta^2 = 0.76$) and memory interval ($F_{(1,9)} = 58.14, p < 0.001, \eta^2 = 0.12$), and their

interaction ($F_{(1,9)} = 10.66, p = 0.01, \eta^2 = 0.02$) on response error. Moreover, conducting paired t-tests within each set size revealed recall error increased with the delay with set size 1 ($t_{(9)} = 5.83, p < .001, d = 1.84$) and set size 3 ($t_{(9)} = 5.78, p < 0.001, d = 1.83$). The interaction effect was a consequence of a larger increase in error with delay for set size 3 compared to set size 1 ($\Delta\mathrm{RMSE} = \mathrm{RMSE}_{7000\mathrm{ms}} - \mathrm{RMSE}_{1000\mathrm{ms}}$). These results are consistent with Experiment 1, corroborating our finding that increasing the set size and delay time have a disadvantageous effect on memory fidelity.

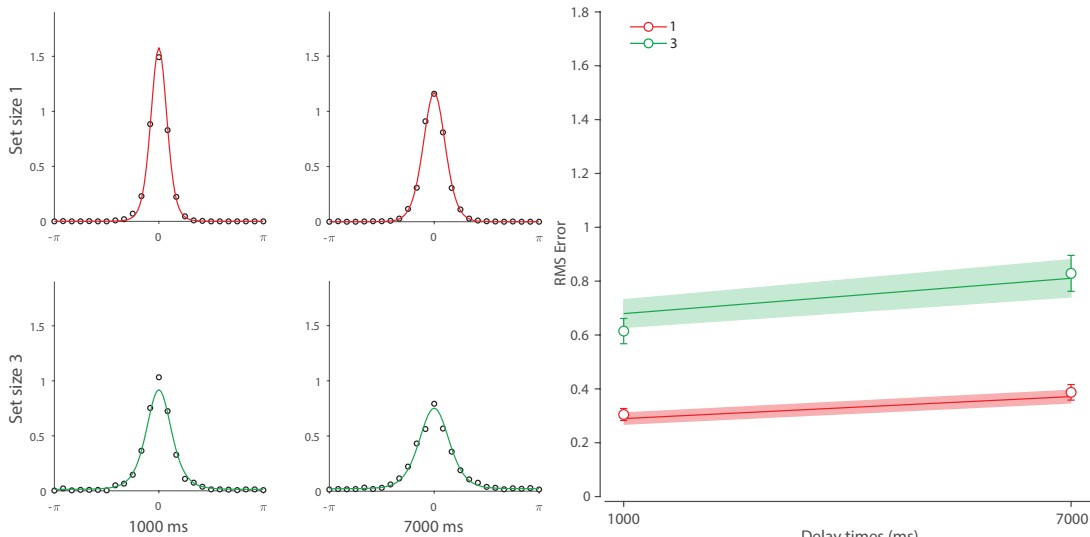

**Appendix 4—figure 2.** Behavioral data and model fit for Experiment 1a. Error bars and patches indicate ±1 SEM. N = 10.

## Neural model

We fitted the DyNR model to the data to test whether noise-driven diffusion is sufficient to account for changes in recall fidelity with longer memory intervals. We applied a simplified version of the model without sensory decay and VWM accumulation components. This was justified given that estimate of sensory decay from Experiment 1 was shorter (mean life $\tau = 0.21$) than the shortest interval used in this experiment (1 s). Moreover, based on our findings in Experiment 2, we argue that a display duration of 500 ms is sufficient to fully encode objects into VWM.

Curves in *Appendix 4—figure 2* show fits of the model with ML parameters (mean ± SE: population gain $\gamma = 385.02 \pm 208.3$, tuning width $\kappa = 2.67 \pm 0.43$, cue processing constant $b = 0.68 \pm 0.67$, base diffusion $\sigma^2_{\mathrm{diff}} = 0.009 \pm 0.001$, swap probability p = 0.005 ± 0.002). The model provided an excellent quantitative fit to response distributions and summary statistics (*Appendix 4—figure 2*), successfully explaining the adverse effects of set size and memory interval on recall fidelity. Critically, and consistent with results from Experiment 1, the proposed DyNR model provided a better fit to human response error compared to the matching model without diffusion (ΔAIC = 144.75) or the model in which diffusion rate increases with set size (ΔAIC = 42.3). In conclusion, this result shows that variability in representations over longer memory intervals can be fully accounted for by noise-driven accumulation without changes in the representational signal (*Schneegans and Bays, 2018*; *Panichello et al., 2019*; *Wolff et al., 2020*).

## Appendix 5

### Additional dataset 2

To further validate predictions of the DyNR model we fitted it to an existing WM study (Experiment 1 in *Bays et al., 2011*). This study focused on the role of temporal dynamics during WM encoding, thereby addressing the same question as our Experiment 2. In contrast to our Experiment 2, *Bays et al., 2011*, used a longer delay period (1100 ms), precluding the strengthening influence of decaying sensory information on recall. This dataset therefore isolates the initial information accumulation process during stimuli presentation.

### Methods

The observers (*N*=32) performed a continuous report task in which a variable number of oriented bars was presented for a variable duration, followed by a pattern mask (100 ms) and a 1 s delay period after which one of the items was probed for recall. Set size was manipulated between observers and exposure duration was manipulated within observers. Each observer performed 100 trials per exposure duration, for a total of 25,600 trials in the study. A more detailed description of the experiment is provided in *Bays et al., 2011*.

### Analysis

Considering only exposure duration in this experiment was manipulated at the observer level, we decided to expand our modeling approach by employing a Bayesian hierarchical method as a compromise between fitting the data for each observer (i.e. set size) independently and pooling the data across all observers. Using a Bayesian hierarchical modeling, individual-observer parameters are considered samples from population distributions, whose means and variances are estimated based on all available data. In general, this approach has a desirable characteristic of constraining individual-level parameters with the population-level distribution and producing meaningful parameter estimates when a model is fitted across separate groups. The dynamic neural model fitted to the data is identical to the model fitted in Experiment 2, with the exception that here we assumed any existing post-stimulus sensory activity completely diminished by the time of the cue (1100 ms post-stimulus offset), and therefore we did not model sensory decay here. To obtain the hierarchical fit, we used the differential evolution Markov chain algorithm (*Braak, 2006*). All individual-level parameters were samples drawn from normal (i.e. Gaussian) distributions, with corresponding mean and standard deviation being constrained by uniform hyperprior distributions. We collected 240,000 post-warmup samples across 12 chains and computed median and 95% equal-tailed intervals (ETI) of posterior distributions to obtain the group and individual-level parameter estimates. Prior specifications and empirical data for all analyses can be found along with the published code.

### Results

*Appendix 5—figure 1* and *Appendix 5—figure 2* show empirical distributions and summary statistics across all conditions. Similar to Experiment 2, increasing the exposure duration ($F_{(7,196)} = 110.9, p < 0.001, \eta^2 = 0.188$) and decreasing the set size ($F_{(3,28)} = 22.83, p < 0.001, \eta^2 = 0.53$) had beneficial effect on response error. Interaction of exposure duration and set size was significant ($F_{(21,196)} = 3.13, p < 0.001, \eta^2 = 0.02$). Critically, the pattern of memory fidelity dynamics largely matches the pattern observed in Experiment 2, with response errors decreasing rapidly as presentation duration was increased from the minimum duration, saturating at longer durations. This pattern was consistent across all set sizes, which only differed in the absolute error.

These dynamics were accurately predicted by the DyNR model, both at the level of response distributions (curves in *Appendix 5—figure 1*) and summary statistics (curves in *Appendix 5—figure 2*). The parameters used to generate model predictions were obtained by taking the individual observer's posterior medians. We observed the following hyperparameters (median and 95% ETI of hyperposterior): population gain $\gamma$ = 109.47 (88.1–133.57), tuning width $\kappa$ = 3.23 (2.6–4.03), sensory rise time constant $\tau_{\text{rise}}$ = 0.0049 (0.0019–0.0091), VWM accumulation time constant $\tau_{\text{WM}}$ = 0.067 ±(0.051–0.087), cue processing constant $b$ = 0.423 (0.093–0.8436), base diffusion $\sigma_{\text{diff}}^2$ = 0.095 (0.057–0.149), spatial uncertainty time constant $\tau_{\text{spatial}}$ = 0.031 (0.022–0.041), swap probability p = 0.02 (0.011–0.034).

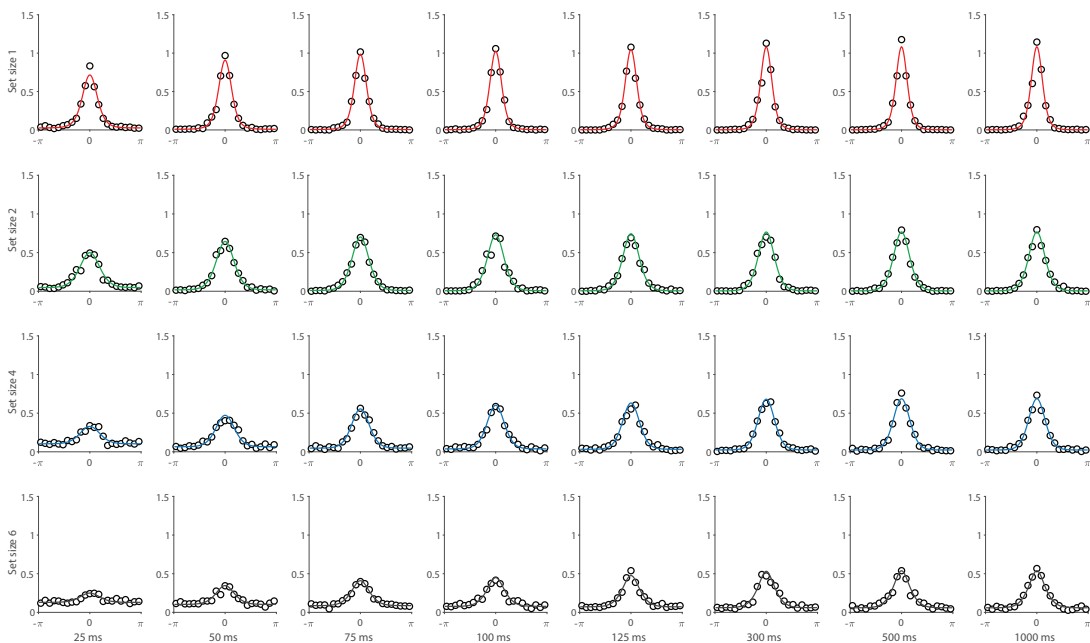

**Appendix 5—figure 1.** Empirical recall error distributions (black circles) from Experiment 1 in *Bays et al., 2011*, and the dynamic neural resource (DyNR) model fits to the data (colored curves).

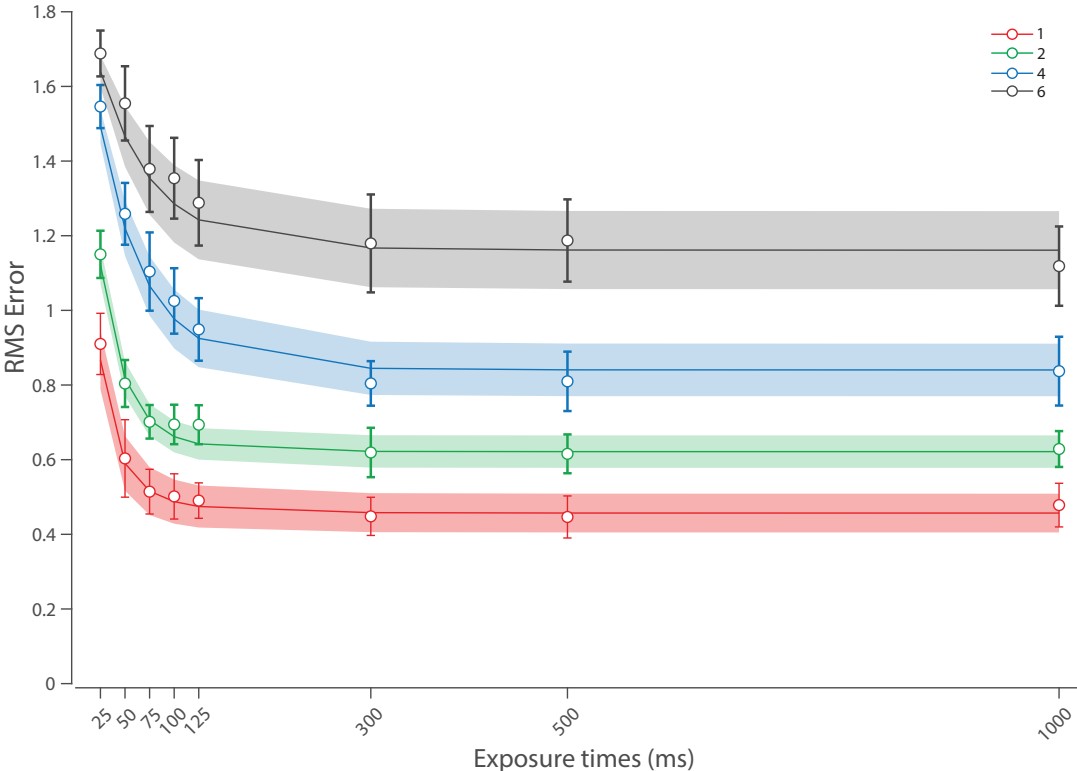

**Appendix 5—figure 2.** Summary statistics (black circles) from Experiment 1 in *Bays et al., 2011* and the dynamic neural resource (DyNR) model fits to the data (colored curves). The DyNR model was fit to the distributions of recall errors shown in *Appendix 5—figure 1*. Error bars and patches indicate ±1 SEM. N = 32.

