## [Editor Report · eLife assessment]

This study presents **important** insights into the dynamical process whereby sensory information is converted from stimulus-driven activity to a working memory representation from which the information can be recalled later. The evidence supporting the claims is **convincing**, using detailed fits and model-comparison techniques applied to new and existing psychophysical data sets to evaluate a wide variety of potential mechanisms. The overall conclusion, that iconic memory and working memory are not distinct mechanisms but rather two slightly different regimes of the same circuitry, will be of interest to neuroscientists and psychologists studying sensory systems and/or working memory.

---

## [Referee Report · Reviewer #2 (Public Review)]

Summary:

Previous work has shown subjects can use a form of short-term sensory memory, known as 'iconic memory', to accurately remember stimuli over short periods of time (several hundred milliseconds). Working memory maintains representations for longer periods of time but is strictly limited in its capacity. While it has long been assumed that sensory information acts as the input to working memory, a process model of this transfer has been missing. To address this, Tomic and Bays present the Dynamic Neural Resource (DyNR) model. The DyNR model captures the dynamics of processing sensory stimuli, transferring that representation into working memory, the diffusion of representations within working memory, and the selection of a memory for report.

The DyNR model captures many of the effects observed in behavior. Most importantly, psychophysical experiments found the greater the delay between stimulus presentation and the selection of an item from working memory, the greater the error. This effect also depended on working memory load. In the model, these effects are captured by the exponential decay of sensory representations (i.e., iconic memory) following the offset of the stimulus. Once the selection cue is presented, residual information in iconic memory can be integrated into working memory, improving the strength of the representation and reducing error. This selection process takes time, and is slower for larger memory loads, explaining the observation that memory seems to decay instantly. The authors compare the DyNR model to several variants, demonstrating the importance of persistence of sensory information in iconic memory, normalization of representations with increasing memory load, and diffusion of memories over time.

Strengths:

The manuscript provides a convincing argument for the interaction of iconic memory and working memory, as captured by the DyNR model. The analyses are cutting-edge and the results are well captured by the DyNR model. In particular, a strength of the manuscript is the comparison of the DyNR model to several alternative variants.

The results provide a process model for interactions between iconic memory and working memory. This will be of interest to neuroscientists and psychologists studying working memory. I see this work as providing a foundation for understanding behavior in continuous working memory tasks, from either a mechanistic, neuroscience, perspective or as a high-water mark for comparison to other psychological process models.

Finally, the manuscript is well written. The DyNR model is nicely described and an intuition for the dynamics of the model are clearly shown in Figures 2 and 4.

Weaknesses:

The manuscript appropriately acknowledges and addresses several minor weaknesses that are due to the limited ability of the approach to disambiguate underlying neural mechanisms. Nevertheless, the manuscript adds significant value to the literature on working memory.

---

## [Referee Report · Reviewer #3 (Public Review)]

Summary

The authors set out to formally contrast several theoretical models of working memory, being particularly interested in comparing the models regarding their ability to explain cueing effects at short cue durations. These benefits are traditionally attributed to the existence of a high capacity, rapidly decaying sensory storage which can be directly read out following short latency retro-cues. Based on the model fits, the authors alternatively suggest that cue-benefits arise from a freeing of working memory resources, which at short cue latencies can be utilized to encode additional sensory information into VWM.

A dynamic neural population model consisting of separate sensory and VWM populations was used to explain temporal VWM fidelity of human behavioral data collected during several working memory tasks. VWM fidelity was probed at several timepoints during encoding, while sensory information was available and maintenance, when sensory information was no longer available. Furthermore, set size and exposure durations were manipulated to disentangle contributions of sensory and visual working memory.

Overall, the model explained human memory fidelity well, accounting for set size, exposure time, retention time, error distributions and swap errors. Crucially the model suggests that recall at short delays is due to post-cue integration of sensory information into VWM as opposed to direct readout from sensory memory. The authors formally address several alternative theories, demonstrating that models with reduced sensory persistence, direct readout from sensory memory, no set-size dependent delays in cue processing and constant accumulation rate provide significantly worse fits to the data.

I congratulate the authors for this rigorous scientific work. All my remarks were thoroughly addressed.

---

## [Author Response]

The following is the authors’ response to the original reviews.

**Reviewer #1 (Recommendations For The Authors):**
I only have a few minor suggestions:Abstract: I really liked the conclusion (that IM and VWM are two temporal extremes of the same process) as articulated in lines 557--563. (It is always satisfying when the distinction between two things that seem fundamentally different vanishes). If something like this but shorter could be included in the Abstract, it would highlight the novel aspects of the results a little more, I think.

Thank you for this comment. We have added the following to the abstract:

“A key conclusion is that differences in capacity classically thought to distinguish IM and VWM are in fact contingent upon a single resource-limited WM store.”

L 216: There's an orphan parenthesis in "(justifying the use)".

Fixed.

L 273: "One surprising result was the observed set size effect in the 0 ms delay condition". In this paragraph, it might be a good idea to remind the reader of the difference between the simultaneous and zero-delay conditions. If I got it right, the results differ between these conditions because it takes some amount of processing time to interpret the cue and free the resources associated with the irrelevant stimuli. Recalling that fact would make this paragraph easier to digest.

That is correct. However, at this point in the text, we have not yet fitted the DyNR model to the data. Therefore, we believe that introducing cue processing and resource reallocation as concepts that differentiate between those two conditions would disrupt the flow of this paragraph. We address these points soon after, in a paragraph starting on line 341.

Figures 3, 5: The labels at the bottom of each column in A would be more clear if placed at the top of each column instead. That way, the x-axis for the plots in A could be labeled appropriately, as "Error in orientation estimate" or something to that effect.

We edited both figures, now Figure 4 and Figure 6, as suggested.

L 379: It should be "(see Eq 6)", I believe.

That is correct, line 379 (currently line 391) should read ‘Eq 6’. Fixed.

L 379--385: I was a bit mystified as to why the scaled diffusion rate produced a worse fit than a constant rate. I imagine the scaled version was set to something likesigma^2_diff_scaled = sigma^2_base + K*(N-1)where N is the set size and sigma^2_base and K are parameters. If this model produced a similar fit as with a constant diffusion rate, the AIC would penalize it because of the extra parameter. But why would the fit be worse (i.e., not match the pattern of variability)? Shouldn't the fitter just find that the K=0 solution is the best? Not a big deal; the Nelder-Mead solutions can wobble when that many parameters are involved, but if there's a simple explanation it might be worth commenting on.

The scaled diffusion was implemented by extending Eq 6 in the following way:

σ(t)2 = (t-toffset) * σ̇ 2diff * N

where N is set size. Therefore, the scaling was not associated with a free parameter that could become 0 if set size did not affect diffusion rate, but variability rather mandatory increased with set size. We now clarify this in the text:

“The second variant was identical to the proposed model, except that we replaced the constant diffusion rate with a set size scaled diffusion rate by multiplying the right side of Eq 6 by N.“

Figure 4 is not mentioned in the main text. Maybe the end of L 398 would be a good place to point to it. The paragraph at L 443-455 would also benefit from a couple of references to it.

Thank you for this suggestion. Figure 4 (now Figure 5) was previously mentioned on line 449 (previously line 437), but now we have included it on line 410 (previously line 398), within the paragraph spanning lines 455-467 (previously 443-455), and also on line 136 where we first discuss masking effects.

L 500: Figure S7 is mentioned before Figures S5 and S6. Quite trivial, I know....

Thank you for this comment. There was no specific reason for Figure S7 to appear after S5 & S6, so we simply swapped their order to be consistent with how they are referred to in the manuscript (i.e., S7 became S5, S5 became S6, and S6 became S7).

**Reviewer #2 (Recommendations For The Authors):**
(1) One potential weakness is that the model assumes sensory information is veridical. However, this isn't likely the case. Acknowledging noise in sensory representations could affect the model interpretation in a couple of different ways. First, neurophysiological recordings have shown normalization affects sensory representations, even when a stimulus is still present on the screen. The DyNR model partially addresses this concern because reports are drawn from working memory, which is normalized. However, if sensory representations were also normalized, then it may improve the model variant where subjects draw directly from sensory representations (an alternative model that is currently described but discarded).

Thank you for this suggestion. We can consider two potential mechanisms through which divisive normalization might be incorporated into sensory processing within the DyNR model.

The first possibility involves assuming that normalization is pre-attentive. In this scenario, the sensory activity of each object would be rescaled at the lowest level of sensory processing, occurring before the allocation of attentional or VWM resources. One strong prediction of such an implementation is that recall error in the simultaneous cue condition (Experiment 1) should vary with set size. However, this prediction is inconsistent with the observed data, which failed to show a significant difference between set sizes, and is more closely aligned with the hypothesis of no-difference (F(2,18) = 1.26, p = .3, η2 = .04, BF10 = 0.47). On that basis, we anticipate that introducing normalization as a pre-attentive mechanism would impair the model fit.

An alternative scenario is to consider normalization as post-attentive. In the simultaneous cueing condition, only one item is attended (i.e., the cued one), regardless of the displayed set size. Here, we would expect normalized activity for a single item, regardless of the number of presented objects, which would then be integrated into VWM. This expanded DyNR model with post-attentive normalization would make exactly the same predictions as the proposed DyNR for recall fidelity, so distinguishing between these models would not be possible based on working memory experiments.

To acknowledge the possibility that sensory signals could undergo divisive normalization and to motivate future research, we have added the following to our manuscript:

“As well as being implicated in higher cognitive processes including VWM (Buschman et al, 2011; Sprague et al., 2014), divisive normalization has been shown to be widespread in basic sensory processing (Bonin et al., 2005; Busse et al., 2009; Ni et al., 2017). The DyNR model presently incorporates the former but not the latter type of normalization. While the data observed in our experiments do not provide evidence for normalization of sensory signals (note comparable recall errors across set size in the simultaneous cue condition of Experiment 1), this may be because sensory suppressive effects are localized and our stimuli were relatively widely separated in the visual field: future research could explore the consequences of sensory normalization for recall from VWM using, e.g., centre-surround stimuli (Bloem et al., 2018).”

Bloem, I. M., Watanabe, Y. L., Kibbe, M. M., & Ling, S. (2018). Visual Memories Bypass Normalization. Psychological Science, 29(5), 845–856. https://doi.org/10.1177/0956797617747091

Bonin, V., Mante, V., & Carandini, M. (2005). The Suppressive Field of Neurons in Lateral Geniculate Nucleus. The Journal of Neuroscience, 25(47), 10844–10856. https://doi.org/10.1523/JNEUROSCI.3562-05.2005

Buschman, T. J., Siegel, M., Roy, J. E., & Miller, E. K. (2011). Neural substrates of cognitive capacity limitations. Proceedings of the National Academy of Sciences, 108(27), 11252–11255. https://doi.org/10.1073/pnas.1104666108

Busse, L., Wade, A. R., & Carandini, M. (2009). Representation of Concurrent Stimuli by Population Activity in Visual Cortex. Neuron, 64(6), 931–942. https://doi.org/10.1016/j.neuron.2009.11.004

Ni, A. M., & Maunsell, J. H. R. (2017). Spatially tuned normalization explains attention modulation variance within neurons. Journal of Neurophysiology, 118(3), 1903–1913. https://doi.org/10.1152/jn.00218.2017

Sprague, T. C., Ester, E. F., & Serences, J. T. (2014). Reconstructions of Information in Visual Spatial Working Memory Degrade with Memory Load. Current Biology, 24(18), 2174–2180. https://doi.org/10.1016/j.cub.2014.07.066

Second, visual adaptation predicts sensory information should decrease over time. This would predict that for long stimulus presentation times, the error would increase. Indeed, this seems to be reflected in Figure 5B. This effect is not captured by the DyNR model.

Indeed, neural responses in the visual cortex have been observed to quickly adapt during stimulus presentation, showing reduced responses to prolonged stimuli after an initial transient (Groen et al., 2022; Sawamura et al., 2006; Zhou et al., 2019). This adaptation typically manifests as (1) reduced activity towards the end of stimulus presentation and (2) a faster decay towards baseline activity after stimulus offset.

In the DyNR model, we use an idealized solution in which we convolve the presented visual signal with a response function (i.e., temporal filter). At the longest presentation durations, in DyNR, the sensory signal plateaus and remains stable until stimulus offset. Because our psychophysical data does not allow us to identify the exact neural coding scheme that underlies the sensory signal, we tend to favour this simple implementation, which is broadly consistent with some previous attempts to model temporal dynamics in sensory responses (e.g., Carandini and Heeger, 1994). However, we agree with the reviewer that some adaptation of the sensory signal with prolonged presentation would also be consistent with our data.

We have added the following to the manuscript:

“In Experiment 2, the longest presentation duration shows an upward trend in error at set sizes 4 and 10. While this falls within the range of measurement error, it is also possible that this is a meaningful pattern arising from visual adaptation of the sensory signal, whereby neural populations reduce their activity after prolonged stimulation. This would mean less residual sensory signal would be available after the cue to supplement VWM activity, predicting a decline in fidelity at higher set sizes. Visual adaptation has previously been successfully accounted for by a type of delayed normalization model in which the sensory signal undergoes a series of linear and nonlinear transformations (Zhou et al., 2019). Such a model could in future be incorporated into DyNR and validated against psychophysical and neural data.”

Carandini, M., & Heeger, D. J. (1994). Summation and division by neurons in primate visual cortex. Science, 264(5163), 1333–1336. https://doi.org/10.1126/science.8191289

Groen, I. I. A., Piantoni, G., Montenegro, S., Flinker, A., Devore, S., Devinsky, O., Doyle, W., Dugan, P., Friedman, D., Ramsey, N. F., Petridou, N., & Winawer, J. (2022). Temporal Dynamics of Neural Responses in Human Visual Cortex. The Journal of Neuroscience, 42(40), 7562–7580. https://doi.org/10.1523/JNEUROSCI.1812-21.2022

Sawamura, H., Orban, G. A., & Vogels, R. (2006). Selectivity of Neuronal Adaptation Does Not Match Response Selectivity: A Single-Cell Study of the fMRI Adaptation Paradigm. Neuron, 49(2), 307–318. https://doi.org/10.1016/j.neuron.2005.11.028

Zhou, J., Benson, N. C., Kay, K., & Winawer, J. (2019). Predicting neuronal dynamics with a delayed gain control model. PLOS Computational Biology, 15(11), e1007484. https://doi.org/10.1371/journal.pcbi.1007484

(2) A second potential weakness is that, in Experiment 1, the authors briefly change the sensory stimulus at the end of the delay (a 'phase shift', Fig. 6A). I believe this is intended to act as a mask. However, I would expect that, in the DyNR model, this should be modeled as a new sensory input (in Experiment 2, 50 ms is plenty of time for the subjects to process the stimuli). One might expect this change to disrupt sensory and memory representations in a very characteristic manner. This seems to make a strong testable hypothesis. Did the authors find evidence for interference from the phase shift?

The phase shift was implemented with the intention of reducing retinal after-effects, essentially acting as a mask for retinal information only; crucially the orientation of the stimulus is unchanged by the phase shift, so from the perspective of the DyNR model, it transmits the same orientation information to working memory as the original stimulus.

If our objective were to model sensory input at the level of individual neurons and their receptive fields, we would indeed need to treat this phase shift as a novel input. Nevertheless, for DyNR, conceived as an idealization of a biological system for encoding orientation information, we can safely assume that visual areas in biological organisms have a sufficient number of phase-sensitive simple cells and phase-indifferent complex cells to maintain the continuity of input to VWM.

When comparing conditions with and without the phase shift of stimuli (Fig S1B), we found performance to be comparable in the perceptual condition (simultaneous presentation) and with the longest delay (1 second), suggesting that the phase shift did not change the visibility or encoding of information into VWM. In contrast, we found strong evidence that observers had access to an additional source of information over intermediate delays when the phase shift was not used. This was evident through enhanced recall performance from 0 ms to 400 ms delay. Based on this, we concluded that the additional source of information available in the absence of a phase shift was accessible immediately following stimulus offset and had a brief duration, aligning with the theoretical concept of retinal afterimages.

(3) It seems odd that the mask does not interrupt sensory processing in Experiment 2. Isn't this the intended purpose of the mask? Should readers interpret this as all masks not being effective in disrupting sensory processing/iconic memory? Or is this specific to the mask used in the experiment?

Visual masks are often described as instantly and completely halting the visual processing of information that preceded the mask. We also anticipated the mask would entirely terminate sensory processing, but our data indicate the effect was not complete (as indicated by model variants in Experiment 2). Nevertheless, we believe we achieved our intended goal with this experiment – we observed a clear modulation of response errors with changing stimulus duration, indicating that the post-stimulus information that survived masking did not compromise the manipulation of stimulus duration. Moreover, the DyNR model successfully accounted for the portion of signal that survived the mask.

We can identify two possible reasons why masking was incomplete. First, it is possible that the continuous report measure used in our experiments is more sensitive than the discrete measures (e.g., forced-choice methods) commonly employed in experiments that found masks to be 100% effective. Second, despite using a flickering white noise mask at full contrast, it is possible that it may not have been the most effective mask; for instance, a mask consisting of many randomly oriented Gabor patches matched in spatial frequency to the stimuli could prove more effective. We decided against such a mask because we were concerned that it could potentially act as a new input to orientation-sensitive neurons, rather than just wiping out any residual sensory activity.

(4) I apologize if I missed it, but the authors did not compare the DyNR model to a model without decaying sensory information for Experiment 1.

We tested two DyNR variants in which the diffusion process was solely responsible for memory fidelity dynamics. These models assumed that the sensory signal terminates abruptly with stimuli offset, and the VWM signal encoding the stimuli was equal to the limit imposed by normalization, independent of the delay duration.

As variants of this model failed to account for the observed response errors both quantitatively (see 'Fixed neural signal' under Model variants) and qualitatively (Figure S3), we decided not to test any more restrictive variants, such as the one without sensory decay and diffusion.

(5) In the current model, selection is considered to be absolute (all or none). However, this need not be the case (previous work argues for graded selection). Could a model where memories are only partially selected, in a manner that is mediated by load, explain the load effects seen in behavior?

Thank you for this point. If attentional selection was partial, it would affect the observers’ efficiency in discarding uncued objects to release allocated resources and encode additional information about the cued item. We and others have previously examined whether humans can efficiently update their VWM when previous items become obsolete. For example, Taylor et al. (2023) showed that observers could efficiently remove uncued items from VWM and reallocate the released resources to new visual information. These findings align with results from other studies (e.g., Ecker, Oberauer, & Lewandowsky, 2014; Kessler & Meiran, 2006; Williams et al., 2013).

Based on these findings, we feel justified in assuming that observers in our current task were capable of fully removing all uncued objects, allowing them to continue the encoding process for the cued orientation that was already partially stored in VWM, such that the attainable limit on representational precision for the cued item equals the maximum precision of VWM.

Partial removal could in principle be modelled in the DyNR model by introducing an additional plateau parameter specifying a maximum attainable precision after the cue. Our concern would be that such a plateau parameter would trade off with the parameter associated with Hick’s law (i.e., cue interpretation time). The former would control the amount of information that can be encoded into VWM, while the latter regulates the amount of sensory information available for encoding. We are wary of adding additional parameters, and hence flexibility, to the model where we do not have the data to sufficiently constrain them.

Ecker, U. K. H., Oberauer, K., & Lewandowsky, S. (2014b). Working memory updating involves item-specific removal. Journal of Memory and Language, 74, 1–15. https://doi.org/10.1016/j.jml. 2014.03.006

Kessler, Y., & Meiran, N. (2006). All updateable objects in working memory are updated whenever any of them are modified: Evidence from the memory updating paradigm. Journal of Experimental Psychology: Learning, Memory, and Cognition, 32, 570–585. https://doi.org/10.1037/0278-7393.32.3.570

Taylor, R., Tomić, I., Aagten-Murphy, D., & Bays, P. M. (2023). Working memory is updated by reallocation of resources from obsolete to new items. Attention, Perception, & Psychophysics, 85(5), 1437–1451. https://doi.org/10.3758/s13414-022-02584-2

Williams, M., & Woodman, G. F. (2012). Directed forgetting and directed remembering in visual working memory. Journal of Experimental Psychology. Learning, Memory, and Cognition, 38(5), 1206–1220. https://doi.org/10.1037/a0027389

(6) Previous work, both from the authors and others, has shown that memories are biased as if they are acted on by attractive/repulsive forces. For example, the memory of an oriented bar is biased away from horizontal and vertical and biased towards diagonals. This is not accounted for in the current model. In particular, this could be one mechanism to generate a non-uniform drift rate over time. As noted in the paper, a non-uniform drift rate could capture many of the behavioral effects reported.

The reviewer is correct that the model does not currently include stimulus-specific effects, although our work on that topic provides a clear template for incorporating them in future (e.g. Taylor & Bays, 2018). Specifically on the question of generating a non-uniform drift, we have another project that currently looks at this exact question (cited in our manuscript as Tomic, Girones, Lengyel, and Bays; in prep.). By examining various datasets with varying memory delays, including the Additional Dataset 1 reported in the Supplementary Information, we found that stimulus-specific effects on orientation recall remain constant with retention time. Specifically, although there is a clear increase in overall error over time, estimation biases remain constant in direction and amplitude, indicating that the bias does not manifest in drift rates (see also Rademaker et al., 2018; Figure S1).

Taylor, R., & Bays, P. M. (2018). Efficient coding in visual working memory accounts for stimulus-specific variations in recall. The Journal of Neuroscience, 1018–18. https://doi.org/10.1523/JNEUROSCI.1018-18.2018

Rademaker, R. L., Park, Y. E., Sack, A. T., & Tong, F. (2018). Evidence of gradual loss of precision for simple features and complex objects in visual working memory. Journal of Experimental Psychology: Human Perception and Performance. https://doi.org/10.1037/xhp0000491

(7) Finally, the authors use AIC to compare many different model variants to the DyNR model. The delta-AICs are high (>10), indicating a strong preference for the DyNR model over the variants. However, the overall quality of fit to the data is not clear. What proportion of the variance in data was the model able to explain? In particular, I think it would be helpful for the reader if the authors reported the variance explained on withheld data (trials, conditions, or subjects).

Thank you for this comment.

Below we report the estimates of r2, representing the goodness of fit between observed data (i.e., RMSE) and the DyNR model predictions.

In Experiment 1, the r2 values between observations and predictions were computed across delays for each set size, yielding the following estimates: r2ss1 = 0.60; r2ss4 = 0.87; r2ss10 = 0.95. Note that lower explained variance for set size 1 arises from both data and model predictions having near-constant precision.

In Experiment 2, we calculated r2 between observations and predictions across presentation durations, separately for each set size, resulting in the following estimates: r2ss1 = 0.88; r2ss4 = 0.71; r2ss10 = 0.70. Note that in this case the decreasing percentage of explained variance with set size is a consequence of having less variability in both data and model predictions with larger set sizes.

While these estimates suggest that the DyNR model effectively fits the psychophysical data, a more rigorous validation approach would involve cross-validation checks across all conditions with a withheld portion of trials. Regrettably, due to the large number of conditions in each experiment, we could only collect 50 trials per condition. We are sceptical that fitting the model to even fewer trials, as necessary for cross-validation, would provide a reliable assessment of model performance.

Minor: It isn't clear to me why the behavioral tasks are shown in Figure 6. They are important for understanding the results and are discussed earlier in the manuscript (before Figure 3). This just required flipping back and forth to understand the task before I could interpret the results.

Thank you for this comment. We have now moved the behavioural task figure to appear early in the manuscript (as Figure 3).

**Reviewer #3 (Recommendations For The Authors):**
(1) Dynamics of sensory signals during perceptionI believe that the modeled sensory signal is a reasonable simplification and different ways to model the decay function are discussed. I would like to ask the authors to discuss the implications of slightly more complex initial sensory transients such as the ones shown in Teeuwen (2021). Specifically for short exposure times, this might be particularly relevant for the model fits as some of the alternative models diverge from the data for short exposures. In addition, the role of feedforward (initial transient?) and feedback signaling (subsequent "plateau" activity) could be discussed. The first one might relate more strongly to sensory signals whereas the latter relates more to top-down attention/recurrent processing/VWM.Particularly, this latter response might also be sensitive to the number of items present on the screen which leads to a related question pertaining to the limitations of attention during perception. Some work suggests that perception is similarly limited in the amount of information that can be represented concurrently (Tsubomi, 2013). Could the authors discuss the implications of this hypothesis? What happens if maximum sensory amplitude is set as a free parameter in the model?Tsubomi, H., Fukuda, K., Watanabe, K., & Vogel, E. K. (2013). Neural limits to representing objects still within view. Journal of Neuroscience, 33(19), 8257-8263.

Thank you for this question. Below, we unpack it and answer it point by point.

While we agree our model of the sensory response is justified as an idealization of the biological reality, we also recognise that recent electrophysiological recordings have illuminated intricacies of neuronal responses within the striate cortex, a critical neural region associated with sensory memory (Teeuwen et al, 2021). Notably, these recordings reveal a more nuanced pattern where neurons exhibit an initial burst of activity succeeded by a lower plateau in firing rate, and stimulus offset elicits a second small burst in the response of some neurons, followed by a gradual decrease in activity after the stimulus disappears (Teeuwen et al, 2021).

In general, asynchronous bursts of activity in individual neurons will tend to average out in the population making little difference to predictions of the DyNR model. Synchronized bursts at stimulus onset could affect predictions for the shortest presentations in Exp 2, however the model appears to capture the data very well without including them. We would be wary of incorporating these phenomena into the model without more clarity on their universality (e.g., how stimulus-dependent they are), their significance at the population level (as opposed to individual neurons), and most importantly, their prominence in visual areas outside striate cortex. Specifically, while Teeuwen et al. (2021) described activity in V1, our model does not make strong assumptions about which visual areas are the source of the sensory input to WM. Based on these uncertainties we believe the idealized sensory response is justified for use in our model.

Next, thank you for the comment on feedforward and feedback signals. We have added the following to our manuscript:

“Following onset of a stimulus, the visual signal ascends through visual areas via a cascade of feedforward connections. This feedforward sweep conveys sensory information that persists during stimulus presentation and briefly after it disappears (Lamme et al., 1998). Simultaneously, reciprocal feedback connections carry higher-order information back towards antecedent cortical areas (Lamme and Roelfsema, 2000). In our psychophysical task, feedback connections likely play a critical role in orienting attention towards the cued item, facilitating the extraction of persisting sensory signals, and potentially signalling continuous information on the available resources for VWM encoding. While our computational study does not address the nature of these feedforward and feedback signals, a challenge for future research is to describe the relative contributions of these signals in mediating transmission of information between sensory and working memory (Semedo et al., 2022).”

Lamme, V. A., Supèr, H., & Spekreijse, H. (1998). Feedforward, horizontal, and feedback processing in the visual cortex. Current Opinion in Neurobiology, 8(4), 529–535. https://doi.org/10.1016/S0959-4388(98)80042-1

Lamme, V. A. F., & Roelfsema, P. R. (2000). The distinct modes of vision offered by feedforward and recurrent processing. Trends in Neurosciences, 23(11), 571–579. https://doi.org/10.1016/S0166-2236(00)01657-X

Semedo, J. D., Jasper, A. I., Zandvakili, A., Krishna, A., Aschner, A., Machens, C. K., Kohn, A., & Yu, B. M. (2022). Feedforward and feedback interactions between visual cortical areas use different population activity patterns. Nature Communications, 13(1), 1099. https://doi.org/10.1038/s41467-022-28552-w

Finally, both you and Reviewer 2 raised a similar interesting question regarding capacity limitations of attention during perception Such a limitation could be modelled by freely estimating sensory amplitude and implementing divisive normalization to that signal, similar to how VWM is constrained. We can consider two potential mechanisms through which divisive normalization might be incorporated into sensory processing within the DyNR model.

The first possibility involves assuming that normalization is pre-attentive. In this scenario, the sensory activity of each object would be rescaled at the lowest level of sensory processing, occurring before the allocation of attentional or VWM resources. One strong prediction of such an implementation is that recall error in the simultaneous cue condition (Experiment 1) should vary with set size. However, this prediction is inconsistent with the observed data, which failed to show a significant difference between set sizes, and is more closely aligned with the hypothesis of no-difference (F(2,18) = 1.26, p = .3, η2 = .04, BF10 = 0.47). On that basis, we anticipate that introducing normalization as a pre-attentive mechanism would impair the model fit.

An alternative scenario is to consider normalization as post-attentive. In the simultaneous cueing condition, only one item is attended (i.e., the cued one), regardless of the displayed set size. Here, we would expect normalized activity for a single item, regardless of the number of presented objects, which would then be integrated into VWM. This expanded DyNR model with post-attentive normalization would make exactly the same predictions as the proposed DyNR for recall fidelity, so distinguishing between these models would not be possible based on working memory experiments.

To acknowledge the possibility that sensory signals could undergo divisive normalization and to motivate future research, we have added the following to our manuscript:

“As well as being implicated in higher cognitive processes including VWM (Buschman et al, 2011; Sprague et al., 2014), divisive normalization has been shown to be widespread in basic sensory processing (Bonin et al., 2005; Busse et al., 2009; Ni et al., 2017). The DyNR model presently incorporates the former but not the latter type of normalization. While the data observed in our experiments do not provide evidence for normalization of sensory signals (note comparable recall errors across set size in the simultaneous cue condition of Experiment 1), this may be because sensory suppressive effects are localized and our stimuli were relatively widely separated in the visual field: future research could explore the consequences of sensory normalization for recall from VWM using, e.g., centre-surround stimuli (Bloem et al., 2018).”

Bloem, I. M., Watanabe, Y. L., Kibbe, M. M., & Ling, S. (2018). Visual Memories Bypass Normalization. Psychological Science, 29(5), 845–856. https://doi.org/10.1177/0956797617747091

Bonin, V., Mante, V., & Carandini, M. (2005). The Suppressive Field of Neurons in Lateral Geniculate Nucleus. The Journal of Neuroscience, 25(47), 10844–10856. https://doi.org/10.1523/JNEUROSCI.3562-05.2005

Buschman, T. J., Siegel, M., Roy, J. E., & Miller, E. K. (2011). Neural substrates of cognitive capacity limitations. Proceedings of the National Academy of Sciences, 108(27), 11252–11255. https://doi.org/10.1073/pnas.1104666108

Busse, L., Wade, A. R., & Carandini, M. (2009). Representation of Concurrent Stimuli by Population Activity in Visual Cortex. Neuron, 64(6), 931–942. https://doi.org/10.1016/j.neuron.2009.11.004

Ni, A. M., & Maunsell, J. H. R. (2017). Spatially tuned normalization explains attention modulation variance within neurons. Journal of Neurophysiology, 118(3), 1903–1913. https://doi.org/10.1152/jn.00218.2017

Sprague, T. C., Ester, E. F., & Serences, J. T. (2014). Reconstructions of Information in Visual Spatial Working Memory Degrade with Memory Load. Current Biology, 24(18), 2174–2180. https://doi.org/10.1016/j.cub.2014.07.066

(2) Effectivity of retro-cues at long delaysCan the authors discuss how cues presented at long delays (>1000 ms) can still lead to increased memory fidelity when sensory signals are likely to have decayed? A list of experimental work demonstrating this can be found in Souza & Oberauer (2016).

Souza, A. S., & Oberauer, K. (2016). In search of the focus of attention in working memory: 13 years of the retro-cue effect. Attention, Perception, & Psychophysics, 78, 1839-1860.

The increased memory fidelity observed with longer delays between memory array offset and cue does not result from integrating available sensory signals into VWM because the sensory signal would have completely decayed by that time. Instead, research so far has indicated several alternative mechanisms that could lead to higher recall precision for cued items, and we can briefly summarize some of them, which are also reviewed in more detail in Souza and Oberauer (2016).

One possibility is that, after a highly predictive retro-cue indicates the to-be-tested item, uncued items can simply be removed from VWM. This could result in decreased interference for the cued item, and consequently higher recall precision. Secondly, the retro-cue could also indicate which item can be selectively attended to, and thereby differentially strengthening it in memory. Furthermore, the retro-cue could allow evidence to accumulate for the target item ahead of decision-making, and this could increase the probability that the correct information will be selected for response. Finally, the retro-cued stimulus could be insulated from interference by subsequent visual input, while the uncued stimuli may remain prone to such interference.

A neural account of this retro-cue effect based on the original neural resource model has been proposed in Bays & Taylor, Cog Psych, 2018. However, as we did not use a retro-cue design in the present experiments, we have decided not to elaborate on this in the manuscript.

(3) Swap errorsI am somewhat surprised by the empirically observed and predicted pattern of swap errors displayed in Figure S2. For set size 10, swap probability does not consistently increase with the duration of the retention interval, although this was predicted by the author's model. At long intervals, swap probability is significantly higher for large compared to small set sizes, which also seems to contrast with the idea of shared, limited VWM resources. Can the authors provide some insight into why the model fails to reproduce part of the behavioral pattern for swap errors? The sentence in line 602 might also need some reconsideration in this regard.

Determining the ground truth for swap errors poses a challenge. The prevailing approach has been to employ a simpler model that estimates swap errors, such as a three-component mixture model, and use those estimates as a proxy for ground truth. However, this method is not without its shortcomings. For example, the variability of swap frequency estimates tends to increase with variability in the report feature dimension (here, orientation). This is due to the increasing overlap of response probability distributions for swap and non-swap responses. Consequently, the discrepancy between any two methods of swap estimation is most noticeable when there is substantial variability in orientation reports (e.g., 10 items and long delay or short exposure).

When modelling swap frequency in the DyNR model, our aim was to provide a parsimonious account of swap errors while implementing similar dynamics in the spatial (cue) feature as in the orientation (report) feature. This parametric description captured the overall pattern of swap frequency with set size and retention and encoding time, but is still only an approximation of the predictions if we fully modelled memory for the conjunction of cue and report features (as in e.g. Schneegans & Bays, 2017; McMaster et al, 2020).

We expanded the existing text in the section ‘Representational dynamics of cue-dimension features’ of our manuscript:

“… Although we did not explicitly model the neural signals representing location, the modelled dynamics in the probability of swap errors were consistent with those of the primary memory feature. We provided a more detailed neural account of swap errors in our earlier works that is theoretically compatible with the DyNR model (McMaster et al., 2020; Schneegans & Bays, 2017).

The DyNR model successfully captured the observed pattern of swap frequencies (intrusion errors). The only notable discrepancy between DyNR and the three-component mixture model (Fig. S2) arises with the largest set size and longest delay, although with considerable interindividual variability. As the variability in report-dimension increases, the estimates of swap frequency become more variable due to the growing overlap between the probability distributions of swap and non-swap responses. This may explain apparent deviations from the modelled swap frequencies with the highest set size and longest delay where orientation response variability was greatest. “

McMaster, J. M. V., Tomić, I., Schneegans, S., & Bays, P. M. (2022). Swap errors in visual working memory are fully explained by cue-feature variability. Cognitive Psychology, 137, 101493. https://doi.org/10.1016/j.cogpsych.2022.101493

Schneegans, S., & Bays, P. M. (2017). Neural Architecture for Feature Binding in Visual Working Memory. The Journal of Neuroscience, 37(14), 3913–3925. https://doi.org/10.1523/JNEUROSCI.3493-16.2017

(4) Direct sensory readoutThe model assumes that readout from sensory memory and from VWM happens with identical efficiency. Currently, we don't know if these two systems are highly overlapping or are fundamentally different in terms of architecture and computation. In the case of the latter, it might be less reasonable to assume that information readout would happen at similar efficiencies, as it is currently assumed in the manuscript. Perhaps the authors could briefly discuss this possibility.

In the direct sensory read-out model, we did not explicitly model the efficiency of readout from either sensory or VWM store. However, the distinctive prediction of this model is that the precision of recall changes exponentially with delay at every set size, including one item. This prediction does not depend on the relative efficiency of readout from sensory and working memory, but only on the principle that direct readout from sensory memory bypasses the capacity limit on working memory. This prediction is inconsistent with the pattern of results observed in Experiment 1, where early cues did not show a beneficial effect on recall error for set size 1. While the proposal raised by the reviewer is intriguing, even if we were to model the process of readout from both the sensory and VWM stores with different efficiencies, the direct read-out model could not account for the near-constant recall error with delay for set size one.

(5) Encoding of distractorsOne of the model assumptions is that, for simultaneous presentations of memory array and cue only the cued feature will be encoded. Previous work has suggested that participants often accidentally encode distractors even when they are cued before memory array onset (Vogel 2005). Given these findings, how reasonable is this assumption in the authors' model?Vogel, E. K., McCollough, A. W., & Machizawa, M. G. (2005). Neural measures reveal individual differences in controlling access to working memory. Nature, 438(7067), 500-503.

Although previous research suggested that observers can misinterpret the pre-cue and encode one of the uncued items, our results argue against this being the case in the current experiment. Such encoding failures would manifest in overall recall error, resulting in a gradient of error with set size, owing to the presence of more adjacent distractors in larger set sizes. However, when we compared recall errors between set sizes in the simultaneous cue condition, we did not find a significant difference between set sizes, and moreover, our results were more likely under the hypothesis of no-difference (F(2,18) = 1.26, p = .3, η2 = .04, BF10 = 0.47). If observers occasionally encoded and reported one of the uncued items in the simultaneous cue condition, those errors were extremely infrequent and did not affect the overall error distributions.